# Selective Mixup Fine-Tuning for Optimizing Non-Decomposable Objectives

**Shrinivas Ramasubramanian**[*1]**, Harsh Rangwani**[*2]**, Sho Takemori**[*3]**, Kunal Samanta**[2]**,
Yuhei Umeda**[3]**, R. Venkatesh Babu**[2]
[1]Fujitsu Research of India, [2]Indian Institute of Science, [3]Fujitsu Limited

## Abstract

The rise in internet usage has led to the generation of massive amounts of data, resulting in the adoption of various supervised and semi-supervised machine learning algorithms, which can effectively utilize the colossal amount of data to train models. However, before deploying these models in the real world, these must be strictly evaluated on performance measures like worst-case recall and satisfy constraints such as fairness. We find that current state-of-the-art empirical techniques offer sub-optimal performance on these practical, non-decomposable performance objectives. On the other hand, the theoretical techniques necessitate training a new model from scratch for each performance objective. To bridge the gap, we propose **SelMix**, a selective mixup-based inexpensive fine-tuning technique for pre-trained models, to optimize for the desired objective. The core idea of our framework is to determine a sampling distribution to perform a mixup of features between samples from particular classes such that it optimizes the given objective. We comprehensively evaluate our technique against the existing empirical and theoretically principled methods on standard benchmark datasets for imbalanced classification. We find that proposed SelMix fine-tuning significantly improves the performance for various practical non-decomposable objectives across benchmarks.

## 1 Introduction

The rise of deep networks has shown great promise by reaching near-perfect performance across computer vision tasks (He et al., 2022; Kolesnikov et al., 2020; Kirillov et al., 2023; Girdhar et al., 2023). It has led to their widespread deployment for practical applications, some of which have critical consequences (Castelvecchi, 2020). Hence, these deployed models must perform robustly across the entire data distribution and not just the majority part. These failure cases are often overlooked when considering only accuracy as our primary performance metric. Therefore, more practical metrics like Recall H-Mean (Sun et al., 2006), Worst-Case (Min) Recall (Narasimhan & Menon, 2021; Mohri et al., 2019), etc., should be used for evaluation. However, optimizing these practical metrics directly for deep networks is challenging as they cannot be expressed as a simple average of a function of label and prediction pairs calculated for each sample (Narasimhan & Menon, 2021). Optimizing such metrics with constraints is termed formally as **Non-Decomposable Objective** Optimization.

In prior works, techniques exist to optimize such non-decomposable objectives, but their scope has mainly been restricted to linear models (Narasimhan et al., 2014; 2015a). Narasimhan & Menon (2021), recently developed consistent logit-adjusted loss functions for optimizing non-decomposable objectives for deep neural networks. After this work in supervised setup, Cost-Sensitive Self-Training (CSST) (Rangwani et al., 2022) extends it to practical semi-supervised learning (SSL) setup, where both unlabeled and labeled data are present. As these techniques optimize non-decomposable objectives like Min-Recall, the *long-tailed (LT) imbalanced datasets* serve as perfect benchmarks for these techniques. However, CSST pre-training on long-tailed data leads to sub-optimal representations and hurts the mean recall of the models (Fig. 1). Further, these methods require re-training the model from scratch to optimize for each non-decomposable objective, which decreases applicability. Practical methods based on empirical insights have been developed to improve the mean performance

---

* denotes Equal Contribution. Code will be available here: github.com/val-iisc/SelMix/.
Correspondence to `shrinivas.ramasubramanian@gmail.com`, `harshr@iisc.ac.in`.

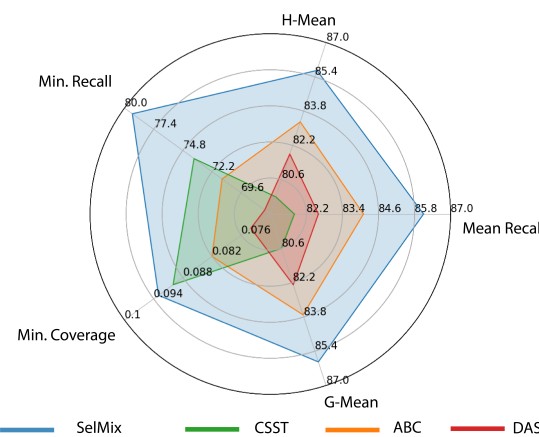

Figure 1: Overview of Results on CIFAR-10 LT (Semi-supervised). We evaluate the models from SotA Semi-supervised techniques of DASO (Oh et al., 2022), ABC (Lee et al., 2021), CSST (Rangwani et al., 2022) and proposed SelMix on different non-decomposable objectives. We find that SelMix produces the best performance for the non-decomposable metric and constraints it is optimized for (blue). Further, SelMix is an inexpensive fine-tuning technique compared to other expensive full pre-training-based baselines.

of methods on long-tailed class-imbalanced datasets (DASO (Oh et al., 2022), ABC (Lee et al., 2021), CoSSL (Fan et al., 2022) etc.). These methods mainly generate debiased pseudo-labels for consistency regularization, leading to better semi-supervised classifiers. Despite their impressive performance, these classifiers perform suboptimally for the non-decomposable objectives (Fig. 1).

In this paper, we develop SelMix, a technique *that utilizes a pre-trained model for representations and optimizes it for improving the desired non-decomposable objective through fine-tuning*. We fine-tune a pre-trained model that provides good representations with Selective Mixups (SelMix) between data across different classes. The core contribution of our work is to develop a selective sampling distribution on class samples to selectively mixup, such that it optimizes the given non-decomposable objective (or metric) (Fig. 2). This SelMix distribution of mixup is updated periodically based on feedback from the validation set so that it steers the model in the direction of optimization of the desired metric. SelMix improves the decision boundaries of particular classes to optimize the objective, unlike standard Mixup (Zhang et al., 2018) that applies mixups uniformly across all class samples. Further, the SelMix framework can also **optimize for non-linear objectives**, addressing a shortcoming of existing works (Rangwani et al., 2022; Narasimhan & Menon, 2021).

To evaluate the performance of SelMix, we perform experiments to optimize several different non-decomposable objectives. These objectives span diverse categories of linear objectives (Min Recall, Mean Recall), non-linear objectives (Recall G-mean, Recall H-mean), and constrained objectives (Recall under Coverage Constraints). We find that the proposed SelMix fine-tuning strategy significantly improves the performance on the desired objective, outperforming both the empirical and theoretical state-of-the-art (SotA) methods in most cases (Fig. 1). In practical scenarios where the distribution of unlabeled data differs from the labeled data, we find that the adaptive design of SelMix with proposed logit-adjusted FixMatch (LA) leads to a significant 5% improvement over the state-of-the-art methods, demonstrating its robustness to data distribution. Further, our SSL framework extends easily to supervised learning and leads to improvement in desired metrics over the existing methods. We summarize our contributions below:

- We evaluate existing theoretical frameworks (Rangwani et al., 2022) and empirical methods (Oh et al., 2022; Wei et al., 2021b; Lee et al., 2021; Fan et al., 2022) on multiple practical non-decomposable metrics. We find that empirical methods perform well on mean recall but poorly on other practical metrics (Fig. 1) and vice-versa for the theoretical method.
- We propose **SelMix**, a mixup-based fine-tuning technique that uses selective mixup over classes to mix up samples to optimize the desired non-decomposable metric objective. (Fig. 2).
- We evaluate SelMix in various supervised and semi-supervised settings, including ones where the unlabeled label distribution differs from that of labeled data. We observe that SelMix with the proposed FixMatch (LA) pre-training significantly outperforms existing SotA methods (Sec. 5).

## 2 RELATED WORKS

**Semi-Supervised Learning in Class Imbalanced Setting.** Semi-Supervised Learning are algorithms that effectively utilizes unlabeled data and the limited labeled data present. A line of work has focused on using consistency regularization and pseudo-label-based self-training on unlabeled data to improve performance (e.g., FixMatch (Sohn et al., 2020), MixMatch (Berthelot et al., 2019b),

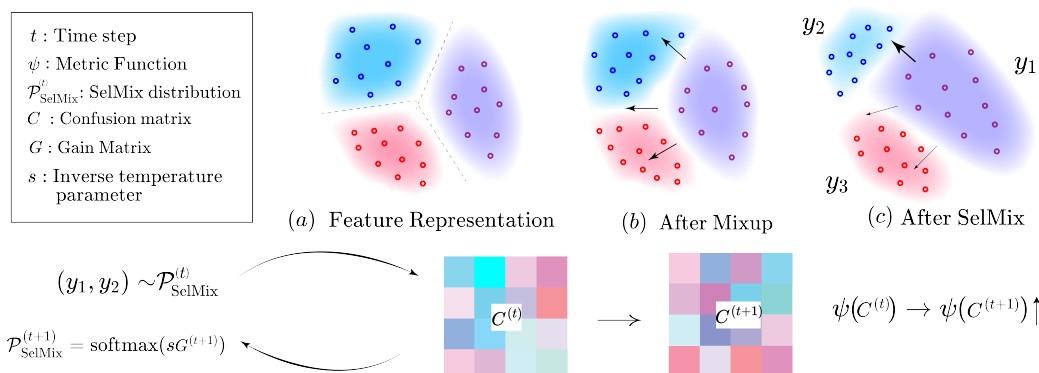

Figure 2: We demonstrate the effect of the variants of mixup on feature representations (a). With Mixup, the feature representation gets equal contribution in all directions of other classes (b). Unlike this, in SelMix (c), certain class mixups are selected at a timestep $t$ such that they optimize the desired metric. Below is an overview of how the SelMix distribution is obtained at timestep $t$.

ReMixMatch (Berthelot et al., 2019a), etc.). However, when naively applied, these methods lead to biased pseudo-labels in class-imbalanced (and long-tailed) settings. Various recent (Fan et al., 2022; Oh et al., 2022) methods have been developed that mitigate pseudo-label bias towards the majority classes. These include an auxiliary classifier trained on balanced data (ABC (Lee et al., 2021)), a semantic similarity-based classifier to de-bias pseudo-label (DASO (Oh et al., 2022)), oversampling minority pseudo-label samples (CReST (Wei et al., 2021a)). Despite their impressive accuracy, these algorithms compromise performance measures focusing on the minority classes.

**Non-Decomposible Metric Optimization.** Despite their impressive accuracy, there still seems to be a wide gap between the performance of majority and minority classes, especially for semi-supervised algorithms. For such cases, suitable metrics like worst-case recall across classes (Mohri et al., 2019) and F-measure (Eban et al., 2017) provide a much better view of the model performance. However, these metrics cannot be expressed as a sum of performance on each sample; hence, they are non-decomposable. Several approaches have been developed to optimize the non-decomposable metrics of interest (Kar et al., 2016; Narasimhan et al., 2014; 2015a; Sanyal et al., 2018; Narasimhan & Menon, 2021). In the recent work of CSST (Rangwani et al., 2022), cost-sensitive learning with logit adjustment has been generalized to a semi-supervised learning setting for deep neural networks. However, these approaches excessively focus on optimizing desired non-decomposable metrics, leading to a drop in the model's average performance (mean recall).

**Variants of MixUp.** After MixUp, several variants of mixup have been proposed in literature like CutMix (Yun et al., 2019), PuzzleMix (Kim et al., 2020b), TransMix (Chen et al., 2022), SaliencyMix (Uddin et al., 2020), AutoMix (Zhu et al., 2020) etc. However, all these methods have focused on creating mixed-up samples, whereas in our work SelMix, we concentrate samples from which classes $(y, y')$ are mixed up to improve the desired metric. Hence, it is complementary to others and can be combined with them. We also provide further discussion in App. N.

## 3 PROBLEM SETUP

**Notation:** For two matrices $A, B \in \mathbb{R}^{m \times n}$ with the same size, we define an inner product $\langle A, B \rangle$ as $\langle A, B \rangle = \text{Tr} \, AB^\top = \sum_{i=1}^{m} \sum_{j=1}^{n} A_{ij} B_{ij}$. For a general function $f : \mathbb{R}^{m \times n} \to \mathbb{R}$ for a matrix variable $X \in \mathbb{R}^{m \times n}$, the directional derivative w.r.t $V \in \mathbb{R}^{m \times n}$ is defined as:

$$\nabla_V f(X) = \lim_{\eta \to 0} \frac{f(X + \eta V) - f(X)}{\eta} \tag{1}$$

which implies $f(X + \eta V) \approx f(X) + \eta \nabla_V f(X)$ for small $\eta$. In case the function $f$ is differentiable, we depict the gradient (derivative) matrix w.r.t $X \in \mathbb{R}^{m \times n}$ as $\frac{\partial f}{\partial X} \in \mathbb{R}^{m \times n}$ with each entry given as: $(\frac{\partial f}{\partial X})_{ij} = \frac{\partial f}{\partial X_{ij}}$. For a comprehensive list of variable definitions used, please refer to Table. A.1.

Let's examine the classification problem involving $K$ classes, where the data points denoted as $x$ are drawn from the instance space $\mathcal{X}$, and the labels belong to the set $\mathcal{Y}$ defined as $[K]$. In this classification task, we define a classifier $F$ as a function that maps

data points to labels. This function is constructed using a neural network-based scoring function $h$, which consists of two parts: a feature extractor $g$ that maps instances $x$ to a feature space $\mathbb{R}^d$ and a linear layer parameterized by weights $W \in \mathbb{R}^{d \times K}$. The classification decision is made by selecting the label that corresponds to the highest score in the output vector: $F(x) = \operatorname{argmax}_{i \in [K]} h(x)_i$. The output of this scoring function $h(x)$ in $\mathbb{R}^K$ is expressed as $h(x) = W^\top g(x)$, in terms of network parameters. We assume access to samples from the data distribution $\mathcal{D}$ for training and evaluation. We denote the prior distribution over labels as $\pi$, where $\pi_i = \mathbf{P}(y = i)$ for $i = 1, \ldots, K$. Now, let's introduce the concept of the

Table 1: Objectives defined by confusion matrix entries.

| Objective | Definition |
|---|---|
| ($\psi^{\mathrm{AM}}$) Mean Recall | $\frac{1}{K} \sum_{i \in [K]} \frac{C_{ii}[h]}{\sum_{j \in [K]} C_{ij}[h]}$ |
| ($\psi^{\mathrm{MR}}$) Min. Recall | $\min_{\lambda \in \Delta_{K-1}} \sum_{i \in [K]} \lambda_i \frac{C_{ii}[h]}{\sum_{j \in [K]} C_{ij}[h]}$ |
| ($\psi^{\mathrm{HM}}$) H-mean | $K \left( \sum_{i \in [K]} \frac{\sum_{j \in [K]} C_{ij}[h]}{C_{ii}[h]} \right)^{-1}$ |
| ($\psi^{\mathrm{AM}}_{\mathrm{cons.}}$) Mean Recall s.t. per class coverage $\geq \tau$ | $\min_{\lambda \in \mathbb{R}^K_+} \sum_{i \in [K]} \frac{C_{ii}[h]}{\sum_{j \in [K]} C_{ij}[h]} + \sum_{j \in [K]} \lambda_j \left( \sum_{i \in [K]} C_{ij}[h] - \tau \right)$ |

confusion matrix, denoted as $C[h]$, which is a key tool for assessing the performance of a classifier, defined as $C_{ij}[h] = \mathbf{E}_{x,y \sim \mathcal{D}}[\mathbb{1}(y = i, \operatorname{argmax}_l h(x)_l = j)]$. For brevity, we introduce the confusion matrix in terms of scoring function $h$. The confusion matrix characterizes how well the classifier assigns instances to their correct classes. An objective function $\psi$ is termed "decomposable" if it can be expressed as a function $\Phi : \mathcal{Y} \times \mathcal{Y} \to \mathbb{R}$. Specifically, $\psi$ is decomposable if it can be written as $\mathbb{E}_{x,y}[\Phi(F(x), y)]$. If such a function $\Phi$ doesn't exist, the objective is termed "non-decomposable". In this context, we introduce the non-decomposable objective $\psi$, represented as $\psi : \Delta_{K \times K-1} \to \mathbb{R}$, which is a function on the set of confusion matrices $C[h]$, and expressed as $\psi(C[h])$. Our primary aim is to maximize this objective $\psi(C[h])$ which can be used to express various practical objectives found in prior research works (Cotter et al., 2019; Narasimhan et al., 2022), examples of which are provided in Table 1 and their real-world usage is described below.

In real-world datasets, a common challenge arises from the inherent long-tailed and imbalanced nature of the distribution of data. In such scenarios, relying solely on accuracy can lead to a deceptive assessment of the classifier. This is because a model may excel in accurately classifying majority classes but fall short when dealing with minority ones. To address this issue effectively, holistic evaluation metrics like H-mean (Kennedy et al., 2010), G-mean (Wang & Yao, 2012; Lee et al., 2021), and Minimum (worst-case) recall (Narasimhan & Menon, 2021) prove to be more suitable. These metrics offer a comprehensive perspective, highlighting performance disparities between majority and minority classes that accuracy might overlook. Specifically, the G-mean of recall can be expressed in terms of the confusion matrix $(C[h])$ as: $\psi^{\mathrm{GM}}(C[h]) = \left( \prod_{i \in [K]} \frac{C_{ii}[h]}{\sum_{j \in [K]} C_{ij}[h]} \right)^{\frac{1}{K}}$. For the minimum recall ($\psi^{\mathrm{MR}}$), we use the continuous relaxation as used by (Narasimhan & Menon, 2021). By writing the overall objective as min-max optimization over $\lambda \in \Delta_{K-1}$, we have $\max_h \psi^{\mathrm{MR}}(C[h]) = \max_h \min_{\lambda \in \Delta_{K-1}} \sum_{i \in [K]} \lambda_i \frac{C_{ii}[h]}{\sum_{j \in [K]} C_{ij}[h]}$. Fairness is another area where such complex objectives are beneficial. For example, prior works (Cotter et al., 2019; Goh et al., 2016) on fairness consider optimizing the mean recall while constraining the predictive coverage ($\mathrm{cov}_i[C[h]] = \sum_j C_{ji}$) that is the proportion of class $i$ predictions on test data given as $\max_h \frac{1}{K} \sum_{i=1}^K \mathrm{rec}_i[h]$ s.t. $\mathrm{cov}_i[h] \geq \frac{\alpha}{K}$ $\forall i \in [K]$. Optimization of the above-constrained objectives is possible by using the Lagrange Multipliers ($\lambda \in \mathbb{R}^K_{\geq 0}$) as done in Sec. 2 of Narasimhan & Menon (2021). By expressing this above expression in terms of $C[h]$ and through linear approximation, the constrained objective $\psi_{\mathrm{cons.}}(C[h])$ can be considered as: $\max_h \psi^{\mathrm{AM}}_{\mathrm{cons.}}(C[h]) = \max_h \min_{\lambda \in \mathbb{R}^K_{\geq 0}} \frac{1}{K} \sum_{i \in [K]} \frac{C_{ii}[h]}{\sum_{j \in [K]} C_{ij}[h]} + \sum_{j \in [K]} \lambda_j \left( \sum_{i \in [K]} C_{ij}[h] - \frac{\alpha}{K} \right)$. The $\lambda$ for calculating the value of $\psi_{\mathrm{cons.}}(C[h])$ and $\psi^{\mathrm{MR}}(C[h])$, is periodically updated using exponentiated or projected gradient descent as done in (Narasimhan & Menon, 2021). We summarize $\psi(C[h])$ for all non-decomposable objectives we consider in this paper in Table 1. Unlike existing frameworks (Narasimhan & Menon, 2021; Rangwani et al., 2022) in addition to objectives that are linear functions of $C[h]$, we can also optimize for non-linear functions like (G-mean and H-mean) for neural networks.

# 4 SELECTIVE MIXUP FOR OPTIMIZING NON-DECOMPOSABLE OBJECTIVES

This section introduces the proposed SelMix (Selective Mixup) procedure for optimizing desired non-decomposable objective $\psi$. We **a)** first introduce a notion of selective $(\mathbf{i}, \mathbf{j})$ mixup (Zhang et al., 2018) between samples of class $i$ and $j$, **b)** we then define the change (i.e., Gain) in desired objective $\psi$ produced due to $(\mathbf{i}, \mathbf{j})$ mixup, **c)** to prefer $(\mathbf{i}, \mathbf{j})$ mixups which lead to large Gain in objective $\psi$, we introduce a distribution $\mathcal{P}_{\text{SelMix}}$ from which we sample $(\mathbf{i}, \mathbf{j})$ to form training batches **d)** we then conclude by introducing a tractable approximation of change (i.e., gain) in metric for the network $W^{\mathrm{T}}g$, **e)** we summarize the SelMix procedure by providing the Algorithm for training. We provide theoretical results for the optimality of the SelMix procedure in Sec. 4.1. We elucidate each part of SelMix in detail below and provide an overview of the proposed algorithm in Fig. 2.

**Feature Mixup.** In this work, we aim to optimize non-decomposable objectives with a framework utilizing mixup (Zhang et al., 2018). Mixup minimizes the risk for a linear combination of samples and labels (i.e. $(x, y)$ and $(x', y')$) to achieve better generalization. Manifold Mixup (Verma et al., 2019) extends this idea to have **mixups in feature space, which we use in our work**. However, in vanilla mixup, the samples for mixing up are chosen randomly over the classes. This may be useful in the case of accuracy but can be sub-optimal when we aim to optimize for specific non-decomposable objectives (Table K.1) that may require a specific selection of classes in the mixup. Hence, this work focuses on selective mixups between classes and uses them to optimize non-decomposable objectives. Consider a $K$ class classification dataset $D$ containing sample pairs $(x, y)$. For convenience of notation, we denote the subset of instances with a particular label $y = i$ as $D_i = \{x \mid (x, y) \in D, y = i\}$. For the unlabelled part of dataset $\tilde{D}$ containing $x$, we generate these subsets $\tilde{D}_i = \{x' \in \tilde{D} \mid i = \operatorname{argmax}_l h(x)_l\}$ based on the pseudo label $y' = \operatorname{argmax}_l h(x)_l$ from the model $h$. In addition to these training data, we also assume access to $D^{\text{val}}$ containing $(x, y)$. Following semi-supervised mixup framework (Fan et al., 2022) in our work, the mixup between a labeled and pseudo labeled pair of samples (i.e. $(x, y)$ from $D$ and $x'$ from $\tilde{D}$), the features $g(x)$ and $g(x')$ are mixed up, while the label is kept as $y$. The mixup loss for our model with feature extractor $g$ followed by a linear layer with weights $W$ defined as:

$$\mathcal{L}_{\text{mixup}}(g(x), g(x'), y; W) = \mathcal{L}_{\text{SCE}}(W^{\mathrm{T}}(\beta g(x) + (1 - \beta)g(x')), y).$$

Here $\beta \sim U[\beta_{min}, 1], \beta_{min} \in [0, 1]$ and $\mathcal{L}_{\text{SCE}}$ is the softmax cross entropy loss. We define **(i, j) mixups** for classes $i$ and $j$ to be the mixup of samples $x \sim D_i$ and $x' \sim \tilde{D}_j$ and minimization of the corresponding $\mathcal{L}_{\text{mixup}}$ via SGD . For analyzing the effect of $(i, j)$ mixups on the model, we use the loss incurred by mixing the centroids of class samples ($z_i$ and $z_j$), which are defined as $z_k = \mathbb{E}_{x \sim D_k^{\text{val}}}[g(x)]$ for each class $k$. This representative of the expected loss due to $(i, j)$ mixup can be expressed as:

$$\mathcal{L}_{(i,j)}^{\text{mix}} = \mathcal{L}_{\text{mixup}}(z_i, z_j, i; W) \quad \forall i, j \in [K] \times [K]. \tag{2}$$

**Directional vectors using mixup loss.** We define $K^2$ directions as the derivative of the expected mixup loss for each of the $(i, j)$ mixup respectively w.r.t the weights $W$ as $V_{i,j} = -\partial \mathcal{L}_{(i,j)}^{\text{mix}}/\partial W$. These directions correspond to the small change in weights $W$ upon the minimization of $\mathcal{L}_{(i,j)}^{\text{mix}}$ by stochastic gradient descent. Now we want to calculate the change in the non-decomposable objective $\psi$ along these directions $V_{i,j}$. Assuming the existence of directional derivatives in the span of $K^2$ directions and fixed feature extractor $g$, we can write the following using Taylor Expansion (Eq. (1)):

$$\psi(C[W^{\top}g + \eta V_{i,j}^{\top}g]) = \psi(C[W^{\top}g]) + \eta \nabla_{V_{ij}}\psi(C[W^{\top}g]) + O(\eta^2 \|V_{i,j}\|^2). \tag{3}$$

In Eq. (3), since $\eta$ is a small scalar, $O(\eta^2 \|V_{i,j}\|^2)$ is negligible. Hence the second term in Eq. (3) denotes the major change in objective $\psi$ due to minimization of the loss due to $(i, j)$ mixup $\mathcal{L}_{(i,j)}^{\text{mix}}$ via SGD. We define this as Gain $(G_{ij})$ or increase in desired objective $\psi$ for the $(i, j)$ mixup:

$$G_{ij} = \nabla_{V_{ij}}\psi(C[W^{\top}g]), \quad \text{where } V_{ij} = -\frac{\partial \mathcal{L}_{(i,j)}^{\text{mix}}}{\partial W} \quad \forall (i, j) \in [K] \times [K] \tag{4}$$

Using this, we define the gain matrix as $G = [G_{i,j}]_{1 < i,j < K}$ corresponding to each of the $(i, j)$ mixups. We now define a general direction $V$, which is induced by mixing up samples from classes $(i, j)$ respectively, according to $\mathcal{P}_{Mix}(i, j)$ distribution defined over $[K] \times [K]$. The expected weight change induced by this distribution can be given as (due to linearity of derivatives): $V = \sum_{i,j} \mathcal{P}_{Mix}(i, j)V_{i,j}$.

The change in objective induced by the $\mathcal{P}_{Mix}(i, j)$ distribution can be similarly approximated using the Taylor Expansion for the direction $V = \sum_{i,j} \mathcal{P}_{Mix}(i, j) V_{i,j}$:

$$\psi(C[W^\top g + \eta V^\top g]) = \psi(C[W^\top g]) + \eta \sum_{i,j} \mathcal{P}_{Mix}(i,j) \nabla_{V_{ij}} \psi(C[W^\top g]) + O(\eta^2 \|V\|^2). \quad (5)$$

To maximize change in objective (LHS), we maximize the second term (RHS), as the first term is constant and the third is negligible for a small step $\eta$. On substituting $\nabla_{V_{ij}} \psi(C[W^\top g]) = G_{ij}$, the second term corresponds to $\mathbb{E}[G] = \sum_{i,j} G_{i,j} \mathcal{P}_{Mix}(i,j)$, which is expectation of gain over $\mathcal{P}_{Mix}$. Thus, maximization of the objective is equivalent to finding optimal $\mathcal{P}_{Mix}$ to maximize expected gain $\mathbb{E}[G]$. In practice to maximize objective $\psi$, we will first sample labels to mixup $(y_1, y_2)$ from optimal $\mathcal{P}_{Mix}$ (described below), then will pick $x_1, x_2$ from $D_{y_1}, D_{y_2}$ uniformly to form a batch for training.

**Selective Mixup Sampling Distribution.** In our work, we introduce a novel sampling distribution $\mathcal{P}_{\text{SelMix}}$ for practically optimizing the gain in objective defined as follows:

$$\mathcal{P}_{\text{SelMix}}(i, j) = \text{softmax}(sG)_{ij}. \quad (6)$$

We aim to maximize $\mathbb{E}[G]$. One strategy could be to only mixup samples from classes $(i, j)$ respectively, corresponding to $\max_{i,j} G_{ij}$ or equivalently $s \to \infty$. However, this doesn't work in practice as we do $n$ steps of SGD based on $\mathcal{P}_{\text{SelMix}}$ the linear approximation in Eq. (3) becomes invalid, and later the approximation in Thm. 4.1 (See Table K.1 for empirical evidence). Hence, we select the $\mathcal{P}_{\text{Mix}}$ to be the scaled softmax of the gain matrix as our strategy, with $s > 0$ given as $\mathcal{P}_{\text{SelMix}}$. The proposed $\mathcal{P}_{\text{SelMix}}$ is the distribution which is an intermediate strategy between the random exploratory uniform ($s = 0$) and greedy ($s \to \infty$) strategies which serves as a good sampling function for maximizing gain. We provide theoretical results regarding the optimality of the proposed $\mathcal{P}_{\text{SelMix}}$ in Sec. 4.1.

**Estimation of Gain Matrix.** This notion of gain, albeit accurate, is not practically tractable since $\psi$ is not differentiable w.r.t $W$ in general, as the definition of $C_{ij}[h] = \mathbf{E}_{x,y \sim \mathcal{D}}[\mathbb{1}(y = i, \text{argmax}_l h(x)_l = j)]$ uses a non-smooth indicator function (Sec. C.2). To proceed further with this limitation, we introduce a reformulation of $C$ by defining the $i^{\text{th}}$ row $C_i[h]$ of the confusion matrix in terms of the $i^{\text{th}}$ row $\tilde{C}_i[h]$ of the unconstrained matrix $\tilde{C}[h] \in \mathbb{R}^{K \times K}$ as $C_i[h] = \pi_i \cdot \text{softmax}(\tilde{C}_i[h])$. This reformulation of the confusion matrix $C$ by design satisfies the necessary constraints, given as $\sum_j C_{i,j}[h] = \pi_i$ and $\sum_{i,j} C_{i,j}[h] = 1$ where $0 \leq C_{i,j}[h] \leq 1$. We can now calculate the same objective $\psi(\tilde{C}[W^\text{T} g])$ in terms of $\tilde{C}$. The entries of gain matrix $(G)$ with reformulation $\tilde{C}$ can be analogously defined (Eq. (4)) in terms of $\tilde{C}$:

$$G_{ij} = \nabla_{V_{ij}} \psi(\tilde{C}[W^\top g]), \quad \text{where } V_{ij} = -\frac{\partial \mathcal{L}_{(i,j)}^{\text{mix}}}{\partial W} \quad \forall (i,j) \in [K] \times [K]. \quad (7)$$

The exact computation of $G_{ij}$ would be given as $\langle \frac{\partial \tilde{C}}{\partial W}, V_{ij} \rangle$ in case $\frac{\partial \tilde{C}}{\partial W}$ was defined. However, this is not defined despite introducing the re-formulation due to the non-differentiability of $\tilde{C}$ w.r.t $W$. However, with re-formulation under some mild assumptions, given in Theorem (4.1), we can approximate $G_{ij}$ (first RHS term). We refer readers to Theorem C.3 for a more mathematically precise statement. Further, we want to convey one advantage despite the proposed reformulation: we do not require the actual computation of $\tilde{C}$ for gain calculation in Eq. (8) (first term). As all the terms of $\frac{\partial \psi(\tilde{C})}{\partial \tilde{C}_{lk}}$ which we require, can be computed analytically in terms of $C$, which makes this operation inexpensive. We provide the $\frac{\partial \psi(\tilde{C})}{\partial \tilde{C}_{lk}}$ for all $\psi$ in Appendix Sec. D.

**Theorem 4.1.** *Assume that $\|V_{ij}\|$ is sufficiently small. Then, the gain for the $(i, j)$ mixup $(G_{ij})$ can be approximated using the following expression:*

$$G_{ij} = \sum_{k,l} \frac{\partial \psi(\tilde{C})}{\partial \tilde{C}_{kl}} \left( (V_{ij})_l^\top \cdot z_k \right) + O \left( \varepsilon(\tilde{C}, W) + \|V_{ij}\|^2 \right). \quad (8)$$

*where $z_k = \mathbb{E}_{x \sim D_k^{\text{val}}}[g(x)]$ is the mean of the features of the validation samples belonging to class $k$, used to characterize (i,j) mixups (Eq. (2)). The error term $\varepsilon(\tilde{C}, W)$ does not depend on $V_{ij}$, and under reasonable assumptions we can regard this term small (we refer readers to Sec. C.3).*

In the above theorem formulation, we approximate the change in the entries of the unconstrained confusion matrix $\tilde{C}_{i,j}[h]$ with the change in logits for the classifier $W^{\mathrm{T}}g$ with weight $W$ along the direction $V_{i,j}$ as $(V_{ij})_l^{\top} \cdot z_k$. The most non-trivial assumption of the theorem is that for each $k$, the random vector $V_{ij}^{\top} g(x)$ has a small variance, where $x \sim D_k^{\mathrm{val}}$. Intuitively, if $g$ is a sufficiently good feature extractor, then feature vectors $g(x)$ ($x \sim D_k^{\mathrm{val}}$) are distributed near its mean, hence its linear projections $V_{ij}^{\top} g(x)$ has a small variance. Therefore, it is a natural assumption if $g$ is sufficiently good. Moreover, this approximation works well in practice, as demonstrated empirically in Sec. 5.

**Algorithm for training through $\mathcal{P}_{\mathbf{SelMix}}$.** We provide an algorithm for training a neural network through SelMix. Our algorithm shares a high-level framework as used by (Narasimhan et al., 2015b; 2022). The idea is to perform training cycles, in which one estimates the gain matrix $G$ through a validation set ($D^{(\mathrm{val})}$) and uses it to train the neural network for a few Stochastic Gradient Descent (SGD) steps.

As our expressions of gain are based on a linear classifier, we primarily fine-tune the linear classifier. The backbone is fine-tuned at a lower learning rate $\eta$ for slightly better empirical results (Sec. 5). Formally, we introduce the time-dependent notations for gain ($G^{(t)}$), the classifier $h^{(t)}$, the SelMix distribution $\mathcal{P}_{\mathrm{SelMix}}^{(t)}$, weight-direction change $V_{i,j}^{(t)}$ due to the minimization of $\mathcal{L}_{ij}^{\mathrm{mix}}$. As SelMix is a fine-tuning procedure,

---

**Algorithm 1** Training through SelMix

**for** $t = 1$ **to** $T$ **do**
    Compute $\mathcal{P}_{\mathrm{SelMix}}^{(t)} = \mathrm{softmax}(sG^{(t)})$ using Thm. 4.1
    **for** $n$ SGD steps **do**
        $Y_1, Y_2 \sim \mathcal{P}_{\mathrm{SelMix}}^{(t)}, X_1 \sim \mathcal{U}(D_{Y_1}), X_2 \sim \mathcal{U}(\tilde{D}_{Y_2})$
        $h^{(t+1)} := \mathrm{SGD\text{-}Update}(h^{(t)}, \mathcal{L}_{\mathrm{mixup}}, (X_1, Y_1, X_2))$
    **end for**
**end for**
**return** $h^{(T)}$

---

we assume a feature extractor $g$ and its linear classification layer, pre-trained with an algorithm like FixMatch, to be provided as input. Another important step in our algorithm is the update of the pseudo-labels of the unlabeled set, $\tilde{D}^{(t)}$. The pseudo-labels are updated with predictions from the $h^{(t)}$. The algorithm is summarized in Alg. 1 and detailed in App. Alg. 2, and an overview is provided in Fig. 2. In our practical implementation, we mask out entries of the gain matrix with negative gain before performing the softmax operation (Eq. (6)). We further compare the SelMix framework to others (Rangwani et al., 2022; Narasimhan & Menon, 2021), in the App. E.

## 4.1 THEORETICAL ANALYSIS OF SELMIX

**Convergence Analysis.** For each iteration $t = 1, \ldots, T$, Algorithm 1 updates the parameter $W$ of our network $h = W^T g$ as follows: (a) It selects a mixup pair $(i,j)$ from the distribution $\mathcal{P}_{\mathrm{SelMix}}^{(t)}$, (b) and updates the parameter $W$ by $W^{(t+1)} = W^{(t)} + \eta_t \widetilde{V}^{(t)}$, where $\widetilde{V}^{(t)} = V_{ij}^{(t)}/\|V_{ij}^{(t)}\|$. Here, we consider the normalized directional vector $\widetilde{V}^{(t)}$ instead of $V_{ij}^{(t)}$. We denote by $\mathbb{E}_{t-1}[\cdot]$ the conditional expectation conditioned on randomness with respect to mixup pairs up to time step $t-1$. In the convergence analysis, we make the following assumptions for the analysis. We assume that the objective $\psi$ as a function of $W$ is concave, differentiable and the gradient is $\gamma$-Lipschitz, where $\gamma > 0$ (Wright, 2015; Narasimhan et al., 2022). We assume that there exists a constant $c > 0$ independent of $t$ satisfying the following condition $\mathbb{E}_{t-1}\left[\widetilde{V}^{(t)}\right] \cdot \frac{\partial \psi(W^{(t)})}{\partial W} > c\|\frac{\partial \psi(W^{(t)})}{\partial W}\|$, that is, $\widetilde{V}^{(t)}$ vector has sufficient alignment with the gradient vector to maximize the objective $\psi$ in expectation. Here, we regard $\psi$ as a function of $W$ in $\frac{\partial \psi(W^{(t)})}{\partial W}$. Moreover, we assume that in the optimization process, $\|W^{(t)}\|$ does not diverge, i.e., we assume that for any $t \geq 1$, we have $\|W^{(t)}\| < R$ with a constant $R > 0$. In practice, this can be satisfied by adding $\ell^2$-regularization of $W$ to the optimization. We define a constant $R_0 > 0$ as $R_0 = \|W^*\| + R$, where $W^* = \mathrm{argmax}_W \psi(W)$. Using the above mild assumptions, we have the following result (we provide a proof in Sec. C.1):

**Theorem 4.2.** *For any $t > 1$, we have $\psi(W^*) - \mathbb{E}\left[\psi(W^{(t)})\right] \leq \frac{4\gamma R_0^2}{c^2(t-1)}$, with an appropriate choice of the learning rate $\eta_t$.*

Theorem 4.2 states that the proposed Algorithm 1 leads to convergence to the optimal metric value $\psi(W^*)$ if $\mathbb{E}\left[V_{ij}^{(t)}\right]$ is a reasonable directional vector for optimization of $\psi$.

**Validity of the mixup sampling distribution.** By formalizing the optimization process as an online learning problem (a similar setting to that of Hedge (Freund & Schapire, 1997)), we state that our sampling method is valid. For conciseness, we only provide an informal statement here and refer to Sec. C.4 for a more precise formulation. We suppose that a mixup pair $(i,j)$ is sampled by

Table 2: Comparison of metric values with various Semi-supervised Long-Tailed methods on CIFAR-10/100 LT under $\rho_l = \rho_u$ setup. The best results are indicated in bold.

| | CIFAR-10 ($\rho_l = \rho_u = 100, N_1 = 1500, M_1 = 3000$) | | | | | CIFAR-100 ($\rho_l = \rho_u = 10, N_1 = 150, M_1 = 300$) | | | | |
|---|---|---|---|---|---|---|---|---|---|---|
| | Mean Rec. | Min Rec. | HM | GM | Mean Rec./Min Cov. | Mean Rec. | Min H-T Rec. | HM | GM | Mean Rec./Min H-T Cov. |
| DARP (Kim et al., 2020a) | 83.3±0.4 | 66.4±3.1 | 81.9±0.5 | 82.6±0.4 | 83.3±0.3/0.0040±3e-3 | 56.5±0.2 | 39.6±1.1 | 48.7±1.3 | 55.4±0.5 | 56.5±0.2/0.0040±2e-3 |
| CReST (Wei et al., 2021b) | 82.1±0.6 | 68.2±3.2 | 81.0±0.7 | 81.6±0.7 | 82.1±0.6/0.073±5e-3 | 58.2±0.2 | 40.7±0.7 | 48.3±0.2 | 54.1±0.1 | 58.2±0.2/0.0083±2e-4 |
| CReST+ (Wei et al., 2021b) | 83.1±0.3 | 71.3±1.5 | 82.2±0.2 | 82.6±0.3 | 83.1±0.3/0.076±2e-3 | 57.8±0.8 | 42.1±0.7 | 48.2±0.6 | 53.8±0.9 | 57.8±0.5/0.0088±1e-4 |
| ABC (Lee et al., 2021) | 85.1±0.5 | 74.1±0.6 | 84.6±0.5 | 84.9±0.6 | 85.1±0.5/0.086±3e-3 | 59.7±0.2 | 46.4±0.6 | 50.1±1.2 | 55.6±0.4 | 59.7±0.2/0.0089±3e-4 |
| CoSSL (Fan et al., 2022) | 82.0±0.3 | 70.6±0.9 | 81.3±0.5 | 81.6±0.3 | 82.0±0.3/0.074±4e-3 | 57.9±0.4 | 46.3±0.5 | 53.7±0.8 | 55.2±0.7 | 57.9±0.4/0.0051±3e-4 |
| DASO (Oh et al., 2022) | 84.1±0.3 | 72.6±2.1 | 83.5±0.3 | 83.8±0.3 | 84.1±0.3/0.083±1e-3 | **60.6**±0.2 | 40.9±0.4 | 49.1±0.7 | 55.9±0.1 | 60.6±0.2/0.0063±3e-4 |
| CSST (Rangwani et al., 2022) | 81.1±0.2 | 71.7±0.2 | 76.9±0.2 | 77.7±0.7 | 81.1±0.2/0.090±2e-4 | 57.2±0.2 | 48.4±0.3 | 47.7±0.8 | 53.5±0.4 | 57.2±0.2/**0.0099**±2e-3 |
| FixMatch(LA) | 79.7±0.6 | 55.9±1.9 | 76.7±0.1 | 78.3±0.1 | 79.7±0.6/0.056±3e-3 | 58.8±0.1 | 34.6±0.6 | 45.5±2.1 | 53.4±0.4 | 58.8±0.1/0.0053±1e-5 |
| w/SelMix (Ours) | **85.4**±0.1 | **79.1**±0.1 | **85.1**±0.1 | **85.3**±0.1 | **85.7**±0.2/**0.095**±1e-3 | 59.8±0.2 | **57.8**±0.5 | **53.8**±0.5 | **56.7**±0.4 | **59.6**±0.5/0.0098±5e-5 |

a distribution $\mathcal{P}^{(t)}$ on $[K] \times [K]$ for $t = 1, \ldots, T$. We call a sequence of sampling distributions $\mathcal{P} = (\mathcal{P}^{(t)})_{t=1}^{T}$ a policy, and call a policy $\mathcal{P}$ non-adaptive if $\mathcal{P}^{(t)}$ is the same for all $1 \le t \le T$. For example, if $\mathcal{P}^{(t)}$ is the uniform distribution for any $t$, then $\mathcal{P}$ is non-adaptive. Then, in Sec. C.4, we shall prove the following statement regarding the optimality of $\mathcal{P}_{\text{SelMix}}$:

**Theorem 4.3** (Informal). *The SelMix policy $\mathcal{P}_{SelMix}$ is approximately better than any non-adaptive policy $\mathcal{P}_{\mathbf{na}} = (\mathcal{P})_{t=1}^{T}$ in terms of the average gain if $T$ is sufficiently large.*

## 5 EXPERIMENTS

We demonstrate the effectiveness of SelMix in optimizing various Non-Decomposable objectives across different labeled and unlabeled data distributions. Following conventions for Long-Tail (LT) classification, $N_i$ and $M_i$ represent the number of samples in the $i^{\text{th}}$ class for the labeled and unlabeled sets, respectively. The label distribution is exponential in nature, and the imbalance factor $\rho$ characterizes it. We define it as $\rho_l = N_1/N_K, \rho_u = M_1/M_K$. In our experiments, we consider the LT semi-supervised version for CIFAR-10,100, Imagenet-100, and STL-10 datasets as done by (Fan et al., 2022; Oh et al., 2022; Kim et al., 2020a; Lee et al., 2021; Rangwani et al., 2022). For the experiments on the long-tailed supervised dataset, we consider the Long-Tailed versions of CIFAR-10, 100, and ImageNet-1k. The parameters for the datasets are available in Tab. G.1.

**Training Details:** Our classifier comprises a feature extractor $g : \mathcal{X} \to \mathbb{R}^d$ and a linear layer with weight $W$ (see Sec. 3). In semi-supervised learning, we use the pre-trained Wide ResNet-28-2 (Zagoruyko & Komodakis, 2016) with FixMatch (Sohn et al., 2020), replacing the loss function with the logit adjusted (LA) cross-entropy loss (Menon et al., 2020) for debiased pseudo-labels. Fine-tuning with SelMix (Alg. 1) includes cosine learning rate and SGD optimizer. In supervised learning, we pre-train models with MiSLAS on ResNet-32 for CIFAR-10, CIFAR-100, and ResNet-50 for ImageNet-1k. We freeze batch norm layers and fine-tune the feature extractor with a low learning rate to maintain stable mean feature statistics $z_k$, as per our theoretical findings. Further details and hyperparameters are provided in appendix Table G.1.

**Evaluation Setup.** We evaluate our work against baselines CReST, CReST+ (Wei et al., 2021b), DASO (Oh et al., 2022), DARP (Kim et al., 2020a), and ABC (Lee et al., 2021) in semi-supervised long-tailed learning. We assess the methods based on two sets of metric objectives: a) **Unconstrained objectives**, including G-mean, H-mean, Mean (Arithmetic Mean), and worst-case (Min.) Recall. b) **Constrained objectives**, involving maximizing recalls under coverage constraints. The constraint requires coverage $\ge \frac{0.95}{K}$ for all classes. For CIFAR-100, we optimize Min Head-Tail Recall/Min Head-Tail coverage instead of Min Recall/Coverage due to its small size. The tail corresponds to the least frequent 10 classes, and the head the rest 90 classes. For detailed metric objectives and definitions, refer to Table G.3. We present results as mean and standard deviation across three seeds.

**Matched Label Distributions.** We report results for $\rho_l = \rho_u$, signifying matched labeled and unlabeled class label distributions. SelMix outperforms FixMatch (LA), achieving a 5% Min Recall boost for CIFAR-10 and a 9.8% improvement in Min HT Recall for CIFAR-100. SelMix also excels in mean recall, akin to accuracy. Its strategy starts with tail class enhancement, transitioning to uniform mixups (App. J).We delve into optimizing coverage-constrained objectives ($\text{cov}_i[h] \ge \frac{0.95}{K}$). Initially, we emphasize mean recall with coverage constraints, supported by CSST. However, SelMix, a versatile method, accommodates objectives like H-mean with coverage (App. I). Table 2 reveals that most SotA methods miss minimum coverage values, except CSST and SelMix. SelMix outperforms CSST in mean recall while meeting constraints, as confirmed in our detailed analysis (App. K.1).

**Unknown Label Distributions.** We address the practical scenario where the labeled data's label distribution differs from that of the unlabeled data ($\gamma_l \ne \gamma_u$). We assess two cases: *a) Mismatched Distributions.* We evaluate various techniques on CIFAR-10 with two mismatched unlabeled dis-

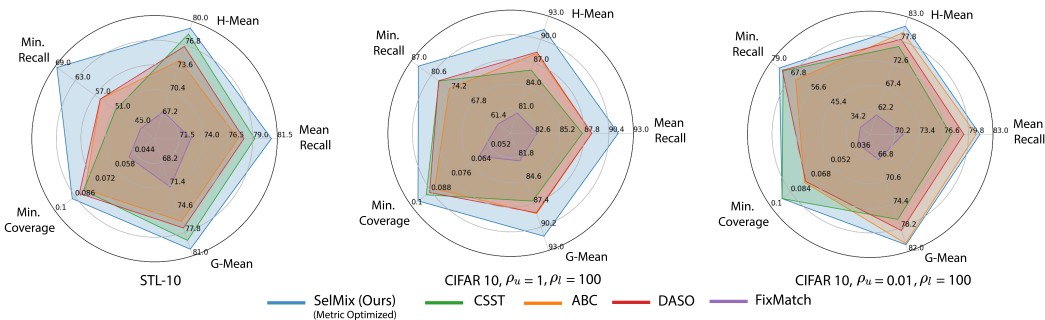

Figure 3: Comparison of metric for semi-supervised CIFAR-10 LT under $\rho_l \neq \rho_u$ and STL-10 $\rho_u = NA$ assumption. For CIFAR-10-LT (semi-supervised) involve $\rho_l = 100$, $\rho_u = 1$, (uniform) and $\rho_l = 100$, $\rho_u = \frac{1}{100}$ (inverted). SelMix achieves significant gains over other baselines.

Table 3: Comparison results in supervised case for CIFAR-10,100 LT ($\rho = 100$). We use the pre-trained model of MiSLAS (Zhong et al., 2021) in stage-1 and fine-tune using SelMix.

| | **CIFAR-10** ($\rho_l = 100$) | | | | | **CIFAR-100** ($\rho_l = 10$) | | | | |
|---|---|---|---|---|---|---|---|---|---|---|
| | Mean Rec. | Min Rec. | HM | GM | Mean Rec./Min Cov. | Mean Rec. | Min H-T Rec. | HM | GM | Mean Rec./Min H-T Cov. |
| MisLaS (Stage 1) (Zhong et al., 2021) | $72.7_{\pm 0.3}$ | $45.6_{\pm 2.3}$ | $70.3_{\pm 1.4}$ | $72.5_{\pm 0.9}$ | $72.7_{\pm 0.3}/0.045_{\pm 2e\text{-}3}$ | $39.5_{\pm 0.2}$ | $1.2_{\pm 0.5}$ | $0.0_{\pm 0.0}$ | $0.0_{\pm 0.0}$ | $39.5_{\pm 0.2}/0.0001_{\pm 2e\text{-}5}$ |
| w/ Stage 2 (Zhong et al., 2021) | $81.9_{\pm 0.1}$ | $72.5_{\pm 0.8}$ | $81.3_{\pm 0.9}$ | $81.6_{\pm 0.1}$ | $81.9_{\pm 0.1}/0.077_{\pm 0.003}$ | $47.0_{\pm 0.4}$ | $15.2_{\pm 1.1}$ | $30.9_{\pm 0.6}$ | $39.9_{\pm 0.5}$ | $47.0_{\pm 0.4}/0.0055_{\pm 2e\text{-}4}$ |
| w/ SelMix (Ours) | $\mathbf{83.3}_{\pm 0.2}$ | $\mathbf{79.2}_{\pm 0.7}$ | $\mathbf{82.6}_{\pm 0.5}$ | $\mathbf{82.8}_{\pm 0.3}$ | $\mathbf{82.8}_{\pm 0.3}/\mathbf{0.095}_{\pm 0.002}$ | $\mathbf{48.3}_{\pm 0.1}$ | $\mathbf{41.3}_{\pm 1.4}$ | $\mathbf{38.2}_{\pm 0.8}$ | $\mathbf{42.3}_{\pm 0.5}$ | $\mathbf{47.8}_{\pm 0.3}/\mathbf{0.0095}_{\pm 2e\text{-}4}$ |

Table 4: Results for scaling SelMix to large datasets ImageNet-1k LT and ImageNet100 LT.

| | **ImageNet100-LT** ($\rho = 10$) | | | | **ImageNet1k-LT** | | |
|---|---|---|---|---|---|---|---|
| | Mean Rec. | Min Rec. | Mean Rec./Min Cov. | | Mean Rec. | Min HT Rec. | Mean Rec./Min Cov. |
| CSST (Rangwani et al., 2022) | 59.1 | 12.1 | 59.1/0.003 | MiSLAS (Stage 1) (Zhong et al., 2021) | 45.4 | 4.1 | 45.4/0.00000 |
| Fixmatch (LA) | 69.9 | 6.0 | 69.9/0.002 | w/ Stage 2 | 52.4 | 29.7 | 52.4/0.00068 |
| w/ SelMix | **73.5** | **24.0** | **73.1/0.009** | w/ SelMix | **52.8** | **45.1** | **52.5/0.00099** |

tributions: balanced ($\rho_u = 1$) and inverse ($\frac{1}{100}$). SelMix consistently outperforms all methods, especially in min. Recall and coverage-constrained objectives (Fig. 3). *b) Real World Unknown Label Distributions.* STL-10 provides an additional 100k samples with an unknown label distribution, emulating scenarios where data is abundant but labels are scarce. SelMix, with no distributional assumptions, outperforms SotA methods like CSST and CRest (which assume matched distribution) in min-recall by a substantial 12.7% margin (Fig. 3). Detailed results can be found in App. H.

**Results on SelMix in Supervised Learning.** To further demonstrate the generality of SelMix, we test it for optimizing non-decomposable objectives via fine-tuning a recent SotA work MisLaS (Zhong et al., 2021) for supervised learning. In comparison to fine-tuning stage-2 of MisLaS, SelMix-based fine-tuning achieves better performance across all objectives as in Table 3, for both CIFAR 10,100-LT.

**Analysis of SelMix.** We demonstrate SelMix's scalability on large-scale datasets like Imagenet-1k LT and Imagenet-100 LT and its ability to improve the objective compared to the baseline Tab. 4, with minimal additional compute cost ($\sim$ 2 min.) (see Table L.1), through Thm. C.8, we show the advantage of SelMix over uniform random sampling and the limitations of a purely greedy policy (ref. Table K.1 for empirical evidence). We observe improved feature extractor learning by comparing a trainable backbone to a frozen one (Tab. K.4). Additionally, our work can be combined with other mixup variants like (Kim et al., 2020b; Yun et al., 2019), leading to performance enhancements (Tab. K.5) demonstrating the diverse applicability for the proposed SelMix method. We refer readers to the Appendix for details on complexity (Sec. B, L), analysis (Sec. K), and limitations (Sec. M).

## 6 CONCLUSION AND DISCUSSION

We study the optimization of practical but complex metrics like the G-mean and H-mean of Recalls, along with objectives with fairness constraints in the case of neural networks. We find that SotA techniques achieve sub-optimal performance in terms of these practical metrics, notably on worst-case recall. These metrics and constraints are non-decomposable objectives, for which we propose a Selective Mixup (SelMix) based fine-tuning algorithm for optimizing them. The algorithm selects samples from particular classes to a mixup to improve a tractable approximation of the non-decomposable objective. Our method, SelMix, can improve on the majority of objectives in comparison to both theoretical and empirical SotA methods, bridging the gap between theory and practice. We expect the SelMix fine-tuning technique to be used for improving existing models by improving on worst-case and fairness metrics inexpensively.

ACKNOWLEDGMENTS

The authors would like to thank Yasunari Hikima for providing useful comments on a preliminary version of this paper. We thank the KIAC Centre for its financial support. Harsh Rangwani is supported by the Prime Minister's Research Fellowship program.

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

APPENDIX

# A NOTATION

We provide a summary of notations in Table A.1.

Table A.1: Table of Notations used in Paper.

| | |
|---|---|
| $K$ | Number of classes |
| $\mathcal{Y} := [K]$ | Label space |
| $x$ | Instance |
| $y$ | Label |
| $\eta$ | Learning rate |
| $\mathcal{X}$ | Instance space |
| $d$ | Feature space |
| $h : \mathcal{X} \to \mathbb{R}^K$ | a neural network based scoring function |
| $C[h]$ | Confusion matrix for the classifier $h$ |
| $\Delta_{n-1} \subset \mathbb{R}^n$ | the $n-1$-dimensional probability simplex |
| $\psi : \Delta_{K^2-1}(\subset \mathbb{R}^{K \times K}) \to \mathbb{R}$ | a function defined on the set of confusion matrices ($\psi(C[h])$ is the metric of $h$) |
| $\pi_i$ | prior for class $i \in [K]$ |
| $g : \mathcal{X} \to \mathbb{R}^d$ | a feature extractor (backbone) |
| $f : \mathbb{R}^d \to \Delta_{K-1}$ | the final classifier such that $h = \text{argmax}_i f_i \circ g$ |
| $W \in \mathbb{R}^{d \times K}$ | the weight of the final layer |
| $z_k$ | the centroid of features of class samples given as $\mathbb{E}_{x \sim D_k^{\text{val}}}[g(x)]$ |
| $\mathcal{L}_{\text{mixup}}(x_i, x_j, y_i; W)$ | the loss for mixup between labeled sample $(x_i, y_i)$ and unlabeled sample $x_j$ |
| $\mathcal{L}_{(ij)}^{\text{mix}}$ | the expected loss due to $(i,j)$ mixup |
| $G_{ij}$ | gain upon performing $(i,j)$ mixup |
| $V_{ij}$ | the directional vector (matrix) defined by the $(i,j)$ mixup |
| $\langle A, B \rangle$ (where $A, B \in \mathbb{R}^{m \times n}$) | $\text{Tr}\, AB^\top$ |
| $\nabla_A \chi$ (where $A \in \mathbb{R}^{d \times K}$) | the directional derivative of a function $\chi$ with the directional vector (matrix) $A$ |
| $s$ | the inverse temperature parameter for the softmax |
| $\tilde{C}$ | Unconstrained extension for confusion matrix $C$ |
| $D_i$ | Subset of data with label $i$ |
| $\tilde{D}_i$ | Subset of data with pseudo-label $i$ |
| $\mathcal{P}$ | a distribution on $[K] \times [K]$ |
| $\boldsymbol{\mathcal{P}} = (\mathcal{P}^t)_{t=1}^T$ | a policy (a sequence of distributions $\mathcal{P}^t$) |
| $\overline{G}(\boldsymbol{\mathcal{P}})$ | the expected average gain of $\boldsymbol{\mathcal{P}}$ |
| $N_k$ | the number of samples in the $k$-th labeled class |
| $M_k$ | the number of samples in the $k$-th unlabeled class |
| $\rho_l$ | the class imbalanced factor of the labeled dataset ($\max_{1 \le i,j \le K} N_i/N_j$) |
| $\rho_u$ | the class imbalanced factor of the unlabeled dataset |
| $\mathcal{H}$ | The set of first 90% classes that contains the majority of samples |
| $\mathcal{T}$ | The set of last 10% classes that contains the minority of samples |
| $\|A\|_F$ | the Frobenius norm of a matrix |

# B COMPUTATIONAL COMPLEXITY

We discuss the computational complexity of SelMix and that of an existing method (Rangwani et al., 2022) for non-decomposable metric optimization in terms of the class number $K$. We note that to the best of our knowledge, CSST (Rangwani et al., 2022) is the only existing method for non-decomposable metric optimization in the SSL setting.

**Proposition B.1.** *The following statements hold:*

1. *In each iteration $t$ in Algorithm 1, computational complexity for $\mathcal{P}_{\mathrm{SelMix}}^{(t)}$ is given as $O(K^3)$.*
2. *In each iteration of CSST (Rangwani et al., 2022), it needs procedure that takes $O(K^3)$ time.*

*Here, the Big-O notation hides sizes of parameters of the network other than $K$ (i.e., the number of rows of $W$) and the size of the validation dataset.*

*Proof.* **1.** Computational complexity for the confusion matrix is given as $O(K^3)$ since there are $K^2$ entries and for each entry, evaluating $h^{(t)}(x)$ takes $O(K)$ time for each validation data $x$. For each $1 \leq k \leq K$, computational complexity for $z_k$ is $O(K)$. We compute $\{\mathrm{softmax}(z_k)\}_{1 \leq k \leq K}$, which takes $O(K^2)$ time. The $(m, l)$-th entry of the matrix $\nu_{ij}$ is given as $-\eta\zeta_m(\delta_{il} - \mathrm{softmax}_i(\zeta))$, where $1 \leq m \leq d$, $1 \leq l \leq K$, and $\zeta = \beta z_i + (1 - \beta)z_j \in \mathbb{R}^d$. Therefore, once we compute $\{\mathrm{softmax}(z_k)\}_{1 \leq k \leq K}$, computational complexity for $\{\nu_{ij}\}_{1 \leq i,j \leq K}$ is $O(K^3)$. For each $1 \leq l \leq K$, we put $v_l = \sum_{k=1}^{K} \frac{\partial \psi(C^{(t)})}{\partial \tilde{C}_{kl}} z_k$. Then computational complexity for $\{v_l\}_{1 \leq l \leq K}$ is $O(K^2)$. Since $G_{ij}^{(t)} = \sum_{l=1}^{K} (\nu_{ij}^{(t)})_l^\top \cdot v_l$ is a sum of $K$ dot products of $d$-dimensional vectors, once we compute $\{v_l\}_l$, computational complexity for $\{G_{ij}^{(t)}\}_{1 \leq i,j \leq K}$ is $O(K^3)$. Thus, computational complexity for $\mathcal{P}_{\mathrm{SelMix}}^{(t)}$ is given as $O(K^3)$.

**2.** In each iteration $t$, CSST needs computation of a confusion matrix at validation dataset. Since there are $K^2$ entries and for each entry, $h^{(t)}(x)$ takes $O(K)$ time for each validation data $x$, computational complexity for the confusion matrix is given as $O(K^3)$. Thus, we have our assertion. $\square$

## C  ADDITIONAL THEORETICAL RESULTS AND PROOFS OMITTED IN THE PAPER

### C.1  CONVERGENCE ANALYSIS

We provide convergence analysis of Algorithm 1. For each iteration $t = 1, \ldots, T$, Algorithm 1 updates parameter $W$ as follows:

1. It selects a mixup pair $(i, j)$ from the distribution $\mathcal{P}_{\mathrm{SelMix}}^{(t)}$.
2. and updates parameter $W$ by $W^{(t+1)} = W^{(t)} + \eta_t \widetilde{V}^{(t)}$, where $\widetilde{V}^{(t)} = V_{ij}^{(t)}/\|V_{ij}^{(t)}\|$.

Here, we consider the normalized directional vector $\widetilde{V}^{(t)}$ instead of $V_{ij}^{(t)}$ and $\| \cdot \|$ denotes the Euclidean norm. We denote by $\mathbb{E}_{t-1}[\cdot]$ the conditional expectation conditioned on randomness with respect to mixup pairs up to time step $t - 1$.

**Assumption C.1.** The function $\psi$ (as a function of $W$) is concave, differentiable, the gradient $\frac{\partial \psi}{\partial W}$ is $\gamma$-Lipschitz, i.e., $\|\frac{\partial \psi}{\partial W} - \frac{\partial \psi}{\partial W'}\| \leq \gamma \|W - W'\|$ where $\gamma > 0$. There exists a constant $c > 0$ independent of $t$ satisfying

$$\mathbb{E}_{t-1}\left[\widetilde{V}^{(t)}\right] \cdot \frac{\nabla \psi(W^{(t)})}{\|\nabla \psi(W^{(t)})\|} > c, \tag{9}$$

where $\nabla \psi(W^{(t)}) = \frac{\partial \psi(W^{(t)})}{\partial W}$, that is, $\widetilde{V}^{(t)}$ vector has sufficient alignment with the gradient. Moreover, we make the following technical assumption. We assume that in the optimization process, $\|W^{(t)}\|$ does not diverge, i.e., we assume that for any $t \geq 1$, we have

$$\|W^{(t)}\| < R$$

with a constant $R > 0$. In practice, this can be satisfied by adding $\ell^2$-regularization of $W$ to the optimization. We define a constant $R_0 > 0$ as $R_0 = \|W^*\| + R$, where $W^* = \mathrm{argmax}_W \psi(W)$. We note that a similar boundedness condition using a level set is assumed by (Wright, 2015).

**Theorem C.2.** *Under the above assumptions and notations, we have the following result. For any $t > 1$, we have*

$$\sup_W \psi(W) - \mathbb{E}\left[\psi(W^{(t)})\right] \leq \frac{4\gamma R_0^2}{c^2(t-1)},$$

*with an appropriate choice of the learning rate $\eta_t$.*

*Proof.* By the Taylor's theorem, for each iteration $t$, there exists $s = s_t \in [0, 1]$ such that $\psi(W^{(t+1)}) = \psi(W^{(t)}) + \eta_t \nabla\psi(\xi) \cdot \widetilde{V}^{(t)}$, where $\xi = W^{(t)} + s\eta_t \widetilde{V}^{(t)}$. We decompose $\eta_t \nabla\psi(\xi) \cdot \widetilde{V}^{(t)}$ as $\eta_t(\nabla\psi(\xi) - \nabla\psi(W^{(t)})) \cdot \widetilde{V}^{(t)} + \eta_t \nabla\psi(W^{(t)}) \cdot \widetilde{V}^{(t)}$.

First, we provide a lower bound of the first term. Since $\nabla\psi$ is Lipschitz, we have $\|\nabla\psi(\xi) - \nabla\psi(W^{(t)})\| \leq \gamma\|\xi - W^{(t)}\| \leq \gamma\eta_t$. Thus, by the Cauchy-Schwartz, we have $\eta_t(\nabla\psi(\xi) - \nabla\psi(W^{(t)})) \cdot \widetilde{V}^{(t)} \geq -\gamma\eta_t^2$. Next, we consider the second term. By the assumption on the cosine similarity (9), we have

$$\eta_t \nabla\psi(W^{(t)}) \cdot \mathbb{E}_{t-1}\left[\widetilde{V}^{(t)}\right] \geq c\eta_t\|\nabla\psi(W^{(t)})\|$$

where $c > 0$ is a constant independent of $t$. Here we note that when taking the expectation $\mathbb{E}_{t-1}[\cdot]$, we can regard $W^{(t)}$ as a non-random variable. Thus, we have

$$\mathbb{E}\left[\psi(W^{(t+1)})\right] = \mathbb{E}\left[\psi(W^{(t)}) + \eta_t \nabla\psi(\xi) \cdot \widetilde{V}^{(t)}\right]$$
$$\geq \mathbb{E}\left[\psi(W^{(t)})\right] - \gamma\eta_t^2 + c\eta_t\mathbb{E}\left[\|\nabla\psi(W^{(t)})\|\right]$$

By letting $\eta_t = \frac{c}{2\gamma}\mathbb{E}\left[\|\nabla\psi(W^{(t)})\|\right]$, we see that

$$\mathbb{E}\left[\psi(W^{(t+1)})\right] \geq \mathbb{E}\left[\psi(W^{(t)})\right] + \frac{c^2}{4\gamma}\left(\mathbb{E}\left[\|\nabla\psi(W^{(t)})\|\right]\right)^2$$

We define $\phi_t = \sup_W \psi(W) - \mathbb{E}\left[\psi(W^{(t)})\right]$. By the above argument, we have

$$\phi_{t+1} \leq \phi_t - \frac{c^2}{4\gamma}\left(\mathbb{E}\left[\|\nabla\psi(W^{(t)})\|\right]\right)^2. \tag{10}$$

Then, we can prove the statement by a similar argument to Theorem 1 in (Wright, 2015) as follows. Let $W^* = argmax_W \psi(W)$. By the concavity of $\psi$ and the definition of $R_0$, we have

$$\psi(W^*) - \psi(W^{(t)}) \leq \|\nabla\psi(W^{(t)})\|\|W^* - W^{(t)}\| \leq \|\nabla\psi(W^{(t)})\|R_0.$$

Therefore, we have

$$\phi_t \leq R_0\mathbb{E}\left[\|\nabla\psi(W^{(t)})\|\right].$$

By this inequality and (10), we have

$$\phi_t - \phi_{t+1} \geq A\phi_t^2,$$

where $A = \frac{c^2}{4\gamma R_0^2}$. Thus, noting that $\phi_{t+1} \leq \phi_t$ holds by (10), we have

$$\frac{1}{\phi_{t+1}} - \frac{1}{\phi_t} = \frac{\phi_t - \phi_{t+1}}{\phi_t \phi_{t+1}} \geq \frac{\phi_t - \phi_{t+1}}{\phi_t^2} \geq A.$$

Therefore, it follows that

$$\frac{1}{\phi_t} \geq \frac{1}{\phi_1} + (t-1)A \geq (t-1)A.$$

This completes the proof. □

## C.2 A FORMAL STATEMENT OF THEOREM 4.1 AND REMARKS ON NON-DIFFERENTIABILITY

We provide a more formal statement of Theorem 4.1 (Theorem C.3) and provide its proof.

**Theorem C.3.** *For a matrix $A \in \mathbb{R}^{n \times m}$, we denote by $\|A\|_{\mathrm{F}}$ the Frobenius norm of $A$. We fix the iteration of the gradient descent and assume that the weight $W$ takes the value $W^{(0)}$ and $\widetilde{C}$ takes the value $\widetilde{C}^{(0)}$.*

*We assume that the following inequality holds for all $k \in [K]$ and $l \in [K]$ uniformly $W \in \mathcal{N}_0$, where $\mathcal{N}_0$ is an open neighbourhood of $W^{(0)}$:*

$$|\mathbb{E}_{x_k} \left[ \mathrm{softmax}_l(W^\top g(x_k)) \right] - \mathrm{softmax}_l(\widetilde{C}_k)| \leq \varepsilon.$$

*We also assume that on $\mathcal{N}_0$, $\widetilde{C}$ can be regarded as a smooth function of $W$ and the Frobenius norm of the Hessian is bounded on $\mathcal{N}_0$. Furthermore, we assume that the following small variance assumption with $\widetilde{\varepsilon} > 0$ for all $k$:*

$$\sum_{m=1}^{K} \mathbb{V}_{x_k} \left[ (W^\top g(x_k))_m \right] \leq \widetilde{\varepsilon}.$$

*Then if $\|\Delta W\|_{\mathrm{F}}$ is sufficiently small, there exist a positive constant $c > 0$ depending only on $K$ with $c = O(\mathrm{poly}(K))$ and a positive constant $c' > 0$ such that the following inequality holds:*

$$\left| G_{ij} - \sum_{k=1}^{K} \frac{\partial \psi}{\partial \widetilde{C}_k} (\Delta W)^\top z_k \right| \leq c \left\| \frac{\partial \psi}{\partial C} \right\|_{\mathrm{F}} (\varepsilon + \widetilde{\varepsilon}) + c'(\|\Delta W\|_{\mathrm{F}}^2 + \|\Delta \widetilde{C}\|_{\mathrm{F}}^2).$$

*Here $\Delta W = \tilde{\nu}_{ij}^t$ and $\widetilde{C}_k$ is a column vector such that the $k$-th row of $\widetilde{C}$ is given as $\widetilde{C}_k$, and we consider Jacobi matrices at $\widetilde{C} = \widetilde{C}^{(0)}$ and the corresponding value of $C$.*

We provide some remarks.

**Independence of the choices of $\widetilde{C}$.** Although the matrix $\widetilde{C}$ is not uniquely determined since the softmax function is not one-to-one, the approximation (the first term of the RHS of (8)) is unique as the derivative of all these are unique. We explain this more in detail. In the approximation formula, we only need jacobian $\partial \psi / \partial \widetilde{C} = \frac{\partial \psi}{\partial C} \frac{\partial C}{\partial \widetilde{C}}$. We note that gradient of the softmax function can also be written as a function of the softmax function (i.e., $\frac{\partial \sigma_l}{\partial \xi_m} = \delta_{lm} \sigma_l(\xi) - \sigma_l(\xi) \sigma_m(\xi)$, where $\sigma_l(\xi) = \mathrm{softmax}_l(\xi)$ for $\xi \in \mathbb{R}^K$, $1 \leq l, m \leq K$). Therefore, the first term of the RHS of eq (8) is uniquely determined even if $\widetilde{C}$ is not uniquely determined.

**Non-smoothness of the indicator functions**. In Theorem C.3, we assume that $C$ and $\widetilde{C}$ as smooth functions $W$, however strictly speaking this assumption does not hold since the indicator functions are not differentiable. Thus, in the definition of $G_{ij}$, we used surrogate functions of the indictor functions. In the following, we provide more detailed explanation. In Eq. (4), we define the gain $G_{ij}$ by a directional derivative of $\psi(C)$ with respect to weight $W$. However, strictly speaking, since the definition of the confusion matrix $C$ involves the indicator function, $\psi(C)$ is not a differentiable function of $W$. Moreover, even if gradients are defined, they vanish because of the definition of the indicator function. In the assumption of Theorem C.3 (a formal version of Theorem 4.1), we assume $\widetilde{C}$ is a smooth function of $W$ and it implies $C$ is a differentiable function of $W$. This assumption can be satisfied if we replace the indicator function by surrogate functions of the indicator functions in the definition of the confusion matrix $C$. More precisely, we replace the definition of $C_{ij}[h] = \pi_i \mathbb{E}_{x \sim P_i} \left[ \mathbb{1}(h(x) = j) \right]$ by $\pi_i \mathbb{E}_{x \sim P_i} \left[ s_j(f(x)) \right]$. Here $h(x) = \mathrm{argmax}_k f_k(x)$ as before, $P_i$ is the class conditional distribution $P(x|y=i)$ and $s_j$ is a surrogate function of $p \mapsto \mathbb{1}(\mathrm{argmax}_i p_i = j)$ satisfying $0 \leq s_j(p) \leq 1$ for any $1 \leq j \leq K$, $p \in \Delta_{K-1}$ and $\sum_{j=1}^{K} s_j(p) = 1$ for any $p \in \Delta_{K-1}$. To compute $G_{ij}$, one can directly use the definition of Eq. (4) with the smoothed confusion matrix using surrogate functions of the indicator function. However, an optimal choice of the surrogate function is unknown. Therefore, in this paper, we introduce an unconstrained confusion matrix $\tilde{C}$ and the approximation formula Theorem 4.1 (Theorem C.3). An advantage of introducing $\widetilde{C}$ and the approximation formula is that the RHS of the approximation formula $\sum_{k,l} \frac{\partial \psi(\bar{C})}{\partial \widetilde{C}_{kl}} \left( (\nu_{ij})_l^\top \cdot z_k \right)$ does not depend on the choice of the surrogate function if we use formulas provided in Sec. D with the original (non-differentiable) definition of $C$ (error terms such as $\varepsilon$ in Theorem C.3 do depend on the surrogate function). Since the optimal choice of the surrogate function is unknown, this gives a reliable approximation.

## C.3  PROOF OF THEOREM C.3

*Proof.* In this proof, to simplify notation, we denote $\mathrm{softmax}(z)$ by $\sigma(z)$ for $z \in \mathbb{R}^K$. In this proof, we fix the iteration of the gradient descent and assume that the weight $W$ takes the value $W^{(0)}$ and $\widetilde{C}$

takes the value $\widetilde{C}^{(0)}$. We assume in an open neighborhood of $W^{(0)}$, we have a smooth correspondence $W \mapsto \widetilde{C}$ and that if the value of $W$ changes from $W_0$ to $W_0 + \Delta W$, then $\widetilde{C}$ changes from $\widetilde{C}_0$ to $\widetilde{C}_0 + \Delta \widetilde{C}$. To prove the theorem, we introduce the following three lemmas. We note that by the assumption of the theorem and Lemma C.5, the assumption (13) of Lemma C.6 can be satisfied with

$$\varepsilon_1 = c''(\varepsilon + \widetilde{\varepsilon}),$$

where $c'' > 0$ is a constant depending only on $K$ with $c'' = O(\mathrm{poly}(K))$. Then by Lemma C.6, there exist constants $c_1 = c_1(K)$ and $c_2 = c_2(K)$ depending on only $K$ with $c_1, c_2 = O(\mathrm{poly}(K))$ such that the following inequality holds for all $k$:

$$\left\| \frac{\partial C}{\partial \widetilde{C}_k}\Big|_{\widetilde{C}_k = \widetilde{C}_k^{(0)}} \left( \Delta \widetilde{C}_k - (\Delta W)^\top z_k \right) \right\|_{\mathrm{F}} \le c_1 \varepsilon_1 + c_2 (\| \Delta \widetilde{C}_k \|_{\mathrm{F}}^2 + \| (\Delta W)^\top z_k \|_{\mathrm{F}}^2). \quad (11)$$

Then, we have the following:

$$\left| \frac{\partial \psi}{\partial \widetilde{C}} \Delta \widetilde{C} - \sum_{k=1}^K \frac{\partial \psi}{\partial \widetilde{C}_k} (\Delta W)^\top z_k \right| = \left| \frac{\partial \psi}{\partial C} \frac{\partial C}{\partial \widetilde{C}} \Delta \widetilde{C} - \sum_{k=1}^K \frac{\partial \psi}{\partial C} \frac{\partial C}{\partial \widetilde{C}_k} (\Delta W)^\top z_k \right|$$

$$= \left| \frac{\partial \psi}{\partial C} \sum_{k=1}^K \frac{\partial C}{\partial \widetilde{C}_k} \Delta \widetilde{C}_k - \sum_{k=1}^K \frac{\partial \psi}{\partial C} \frac{\partial C}{\partial \widetilde{C}_k} (\Delta W)^\top z_k \right|$$

$$\le \left\| \frac{\partial \psi}{\partial C} \right\|_{\mathrm{F}} \left\| \sum_{k=1}^K \frac{\partial C}{\partial \widetilde{C}_k} \left( \Delta \widetilde{C}_k - (\Delta W)^\top z_k \right) \right\|_{\mathrm{F}}$$

Here, by fixing an order on $[K] \times [K]$, we regard $\frac{\partial \psi}{\partial \widetilde{C}}$, $\frac{\partial C}{\partial \widetilde{C}}$ and $\Delta \widetilde{C}$ as a $K^2$-dimensional row vector, a $K^2 \times K^2$-matrix, and a $K^2$-dimensional column vector, respectively. Moreover, we consider Jacobi matrices at $\widetilde{C} = \widetilde{C}^{(0)}$. Then, the assertion of the theorem from this inequality, (11), Lemma C.4. $\quad \square$

**Lemma C.4.** *Under assumptions and notations in the proof of Theorem C.3, there exists a constant $c > 0$ such that*

$$\left| G_{ij} - \frac{\partial \psi}{\partial \widetilde{C}}\Big|_{\widetilde{C} = \widetilde{C}^{(0)}} \Delta \widetilde{C} \right| \le c \| \Delta W \|_{\mathrm{F}}^2.$$

*Proof.* By the assumption of the mapping $W \mapsto \widetilde{C}$ and the Taylor's theorem, there exists $c_1 > 0$ such that

$$\left\| \Delta \widetilde{C} - \left( \frac{\partial \widetilde{C}}{\partial W} \right)\Big|_{W = W_0} \Delta W \right\|_{\mathrm{F}} \le c_1 \| \Delta W \|_{\mathrm{F}}^2. \quad (12)$$

By definition of $G_{ij}$, we have the following:

$$\left| G_{ij} - \frac{\partial \psi}{\partial \widetilde{C}}\Big|_{\widetilde{C} = \widetilde{C}^{(0)}} \Delta \widetilde{C} \right| = \left| \frac{\partial \psi}{\partial W} \Delta W - \frac{\partial \psi}{\partial \widetilde{C}} \Delta \widetilde{C} \right|$$

$$= \left| \frac{\partial \psi}{\partial \widetilde{C}} \frac{\partial \widetilde{C}}{\partial W} \Delta W - \frac{\partial \psi}{\partial \widetilde{C}} \Delta \widetilde{C} \right|$$

$$\le c_1 \left\| \frac{\partial \psi}{\partial \widetilde{C}} \right\|_{\mathrm{F}} \| \Delta W \|_{\mathrm{F}}^2.$$

Here we consider Jacobi matrices at $W = W_0$ and corresponding values. The last inequality follows from the fact that the matrix norm $\| \cdot \|_{\mathrm{F}}$ is sub-multiplicative and Eq. (12). $\quad \square$

**Lemma C.5.** *Under assumptions and notations in the proof of Theorem C.3, there exist a positive constant $c = c(K)$ depending only on $K$ with $c = O(\mathrm{poly}(K))$ such that:*

$$|\mathbb{E}\left[ \sigma_l(W^\top g(x_k)) \right] - \sigma_l(W^\top z_k)| \le c \sum_{m=1}^K \mathbb{V}\left[ (W^\top g(x_k))_m \right],$$

*for any $1 \le k, l \le K$.*

*Proof.* This can be proved by applying the Taylor's theorem to $\sigma_l$. We fix $k, l$ and apply the Taylor's theorem to the function $\xi \mapsto \sigma_l(\xi)$ at $\xi = W^\top z_k = W^\top \mathbb{E}\left[g(x_k)\right]$. Then there exists $\xi_0 \in \mathbb{R}^K$ such that

$$\sigma_l(\xi) = \sigma_l(W^\top z_k) + \left.\frac{\partial \sigma_l}{\partial \xi}\right|_{\xi = W^\top z_k} (\xi - W^\top z_k) + \frac{1}{2}(\xi - W^\top z_k)^\top H_k (\xi - W^\top z_k),$$

where $H_k = \left.\frac{\partial^2 \sigma_l}{\partial \xi^2}\right|_{\xi = \xi_0}$. By noting that $\frac{\partial \sigma_l}{\partial \xi_m} = \delta_{lm}\sigma_l(\xi) - \sigma_l(\xi)\sigma_m(\xi)$ (here $\delta_{lm}$ is the Kronecker's delta), it is easy to see that there exists a constant $c'_l$ depending only on $l$ and $K$ such that $\|H\|_\mathrm{F} < c'_l$ and $c'_l = O(\mathrm{poly}(K))$. By letting $\xi = W^\top g(x_k)$ in the above equation and taking the expectation of the both sides, we obtain the assertion of the lemma with $c = \frac{1}{2}\max_{l \le [K]} c'_l$. $\square$

**Lemma C.6.** *Under assumptions and notations in the proof of Theorem C.3, we assume there exists $\varepsilon_1 > 0$ such that the following inequality holds for all $k$ and $l$ for any $W$ in an open neighborhood of $W^{(0)}$ and corresponding $\widetilde{C}$:*

$$\left|\sigma_l(W^\top z_k) - \sigma_l(\widetilde{C}_k)\right| \le \varepsilon_1. \tag{13}$$

*Furthermore, we assume that $\|(\Delta W)^\top z_k\|_\mathrm{F}$ is sufficiently small for all $k$. Then there exist constants $c_1 = c_1(K)$ and $c_2 = c_2(K)$ depending on only $K$ with $c_1, c_2 = O(\mathrm{poly}(K))$ such that*

$$\left\|\left.\frac{\partial C}{\partial \widetilde{C}_k}\right|_{\widetilde{C}_k = \widetilde{C}_k^{(0)}} \left(\Delta \widetilde{C}_k - (\Delta W)^\top z_k\right)\right\|_\mathrm{F} \le c_1 \varepsilon_1 + c_2(\|\Delta \widetilde{C}_k\|_\mathrm{F}^2 + \|(\Delta W)^\top z_k\|_\mathrm{F}^2).$$

*Here, $\widetilde{C}_k$ (resp. $\Delta\widetilde{C}_k$) is a column vector such that the $k$-th row vector of $\widetilde{C}$ (resp. $\Delta\widetilde{C}$) is given as $\widetilde{C}_k$ (resp. $\Delta\widetilde{C}_k$). Moreover, when defining Jacobi matrices, we regard $C$ as a $K^2$-vector and consider a $K^2 \times K$ Jacobi matrix $\left.\frac{\partial C}{\partial \widetilde{C}_k}\right|_{\widetilde{C}_k = \widetilde{C}_k^{(0)}}$ at $\widetilde{C}_k = \widetilde{C}_k^{(0)}$.*

*Proof.* Since (13) holds all $W$ in an open neighborhood of $W^{(0)}$ and corresponding $\widetilde{C}$, we apply the Taylor's theorem to the function $\xi \mapsto \sigma_l(\xi)$ at $\xi = (W^{(0)})^\top z_k$ and $\xi = \widetilde{C}_k^{(0)}$. Then by (13) and the same argument in the proof of Lemma C.5, we have

$$\left|\left.\frac{\partial \sigma_l}{\partial \xi}\right|_{\xi = \mu_k} \Delta\mu_k - \left.\frac{\partial \sigma_l}{\partial \xi}\right|_{\xi = \widetilde{C}_k^{(0)}} \Delta\widetilde{C}_k\right| \le \varepsilon_1 + c'_2(\|\Delta\mu_k\|_\mathrm{F}^2 + \|\Delta\widetilde{C}_k\|_\mathrm{F}^2),$$

where $\mu_k = (W^{(0)})^\top z_k$, $\Delta\mu_k = (\Delta W)^\top z_k$. Noting that $(\frac{\partial \sigma_l}{\partial \xi})_m$ is given as $\delta_{ml}\sigma_l(\xi) - \sigma_m(\xi)\sigma_l(\xi)$, (13) and the assumption that $\|\Delta\mu_k\|_\mathrm{F}$ is sufficiently small, we see that there exists a constant $c'_1, c'_2 > 0$ depending only on $K$ with $c'_1, c'_2 = O(\mathrm{poly}(K))$ such that the following inequality holds:

$$\left|\left.\frac{\partial \sigma_l}{\partial \xi}\right|_{\xi = \widetilde{C}_k^{(0)}} \left(\Delta\mu_k - \Delta\widetilde{C}_k\right)\right| \le c'_1 \varepsilon_1 + c'_2(\|\Delta\mu_k\|_\mathrm{F}^2 + \|\Delta\widetilde{C}_k\|_\mathrm{F}^2). \tag{14}$$

Next, we consider entries of the $K^2$-vector $\left.\frac{\partial C}{\partial \widetilde{C}_k}\right|_{\widetilde{C}_k = \widetilde{C}_k^{(0)}} (\Delta\widetilde{C}_k - \Delta\mu_k)$. Here as previously mentioned by fixing an order on $[K] \times [K]$, we regard $\frac{\partial C}{\partial \widetilde{C}_k}$ as a $K^2 \times K$-matrix. For $(k, l) \in [K] \times [K]$, by the definition of the mapping $\widetilde{C} \mapsto C$, $(k, l)$-th entry of $\left.\frac{\partial C}{\partial \widetilde{C}_k}\right|_{\widetilde{C}_k = \widetilde{C}_k^{(0)}} (\Delta\widetilde{C}_k - \Delta\mu_k)$ is given as $\pi_k \left.\frac{\partial \sigma_l}{\partial \xi}\right|_{\xi = \widetilde{C}_k^{(0)}} (\Delta\widetilde{C}_k - \Delta\mu_k)$. By (14), we see that there exist constants $c''_1, c''_2$ depending only on $K$ and $c''_1, c''_2 = O(\mathrm{poly}(K))$ such that

$$\left\|\left.\frac{\partial C}{\partial \widetilde{C}_k}\right|_{\widetilde{C}_k = \widetilde{C}_k^{(0)}} \left(\Delta\widetilde{C}_k - (\Delta W)^\top z_k\right)\right\|_\mathrm{F} \le c''_1 \varepsilon_1 + c''_2(\|\Delta\widetilde{C}_k\|_\mathrm{F}^2 + \|(\Delta W)^\top z_k\|_\mathrm{F}^2).$$

Since constants $c''_1, c''_2$ may depend on $(k, l)$ by taking $c_1 = \max_{(k,l)} c''_1$ and $c_2 = \max_{(i,l)} c''_2$, we have the assertion of the lemma. $\square$

## C.4 VALIDITY OF THE MIXUP SAMPLING DISTRIBUTION

In this section, Motivated by Algorithm 1, we consider the following online learning problem and prove validity of our method. For each time step $t = 1, \ldots, T$, an agent selects pairs $(i^{(t)}, j^{(t)}) \in [K] \times [K]$, where random variables $(i^{(t)}, j^{(t)})$ follows a distribution $\mathcal{P}^{(t)}$ on $[K] \times [K]$. We call a sequence of distributions $(\mathcal{P}^{(t)})_{t=1}^{T}$ a policy. For $(i, j) \in [K] \times [K]$ and $1 \leq t \leq T$, we assume that random variable $G_{ij}^{(t)}$ is defined. We regard $G_{ij}^{(t)}$ as the gain in the metric when performing $(i, j)$-mixup at iteration $t$ in Algorithm 1. We assume that $G_{ij}^{(t)}$ is random variable due to randomness of the validation dataset, $X_1, X_2$, and the policy. Furthermore, we assume that when selecting $(i^{(t)}, j^{(t)})$, the agent observes random variables $G_{ij}^{(t)}$ for $(i, j) \in [K] \times [K]$ but cannot observe the true gain defined by $\mathbb{E}\left[G_{ij}^{(t)}\right]$. The average gain $\overline{G}^{(T)}(\mathcal{P})$ of a policy $\mathcal{P} = (\mathcal{P}^{(t)})_{t=1}^{T}$ is defined as $\overline{G}^{(T)}(\mathcal{P}) = \frac{1}{T} \sum_{t=1}^{T} \mathbb{E}\left[G_{i^{(t)} j^{(t)}}^{(t)}\right]$, where $(i^{(t)}, j^{(t)})$ follows the distribution $\mathcal{P}^{(t)}$ and the expectation is taken with respect to the randomness of the policy, validation dataset, $X_1, X_2$. This problem setting is similar to that of Hedge (Freund & Schapire, 1997) (i.e., online convex optimization). However, in the problem setting of Hedge, the agent observes gains (or losses) after performing an action but in our problem setting, the agent have random estimations of the gains before performing an action. We note that even in this setting, methods such as argmax with respect to $G_{ij}^{(t)}$ may not perform well due to randomness of $G_{ij}^{(t)}$ and errors in the approximation (Refer Sec. 5 for evidence).

We call a policy $\mathcal{P} = (\mathcal{P}^{(t)})_{t=1}^{T}$ non-adaptive (or stationary) if $\mathcal{P}^{(t)}$ is the same for all $t = 1, \ldots, T$, i.e, if there exists a distribution $\mathcal{P}^{(0)}$ on $[K] \times [K]$ such that $\mathcal{P}^{(t)} = \mathcal{P}^{(0)}$ for all $t = 1, \ldots, T$. A typical example of non-adaptive policies is the uniform mixup, i.e., $\mathcal{P}^{(t)}$ is the uniform distribution on $[K] \times [K]$. Another example is $\mathcal{P}^{(t)} = \delta_{(i^{(0)}, j^{(0)})}$ for a fixed $(i^{(0)}, j^{(0)}) \in [K] \times [K]$ (i.e., the agent performs the fixed $(i^{(0)}, j^{(0)})$-mixup in each iteration). Similarly to Hedge (Freund & Schapire, 1997) and EXP3 (Auer et al., 2002), we define $\mathcal{P}_{\text{SelMix}} = (\mathcal{P}_{\text{SelMix}}^{(t)})_{t=1}^{T}$ by $\mathcal{P}_{\text{SelMix}}^{(t)} = \text{softmax}((s \sum_{\tau=1}^{t} G_{ij}^{(\tau)})_{1 \leq i,j \leq K})$, where $s > 0$ is the inverse temperature parameter. The following theorem states that $\mathcal{P}_{\text{SelMix}}$ is better than any non-adaptive policy in terms of the average expected gain if $T$ is large:

**Theorem C.7.** *We assume that $G_{ij}^{(t)}$ is normalized so that $|G_{ij}^{(t)}| \leq 1$. Then, for any non-adaptive policy $\mathcal{P}^{(0)} = (\mathcal{P}^{(0)})_{t=1}^{T}$, we have $\overline{G}^{(T)}(\mathcal{P}_{\text{SelMix}}) + \frac{2 \log K}{sT} \geq \overline{G}^{(T)}(\mathcal{P}^{(0)})$.*

*Proof of Theorem C.7.* This can be proved by standard argument of the proof of the mirror descent method (see e.g. (Lattimore & Szepesvári, 2020), chapter 28).

Denote by $\Delta \subset \mathbb{R}^{K \times K}$ the probability simplex of dimension $K^2 - 1$. Let $(i_0, j_0) \in K \times K$ be the best fixed mixup hindsight. Since any non-adaptive policy is no better than the best fixed mixup in terms of $\overline{G}$, we may assume that $\mathcal{P}^{(0)} = (\pi_0)_t$, where $\pi_0$ is the one-hot vector in $\Delta$ defined as $(\pi_0)_{ij} = 1$ if $(i, j) = (i_0, j_0)$ and 0 otherwise for $1 \leq i, j \leq K$. Let $F$ be the negative entropy function, i.e., $F(p) = \sum_{i,j=1}^{K} p_{ij} \log p_{ij}$. For $p \in \Delta$ and $G \in \mathbb{R}^{K \times K}$, we define $\langle p, G \rangle = \sum_{i,j=1}^{K} p_{ij} G_{ij}$. Then, it is easy to see that $p^{(t)} = \mathcal{P}_{\text{SelMix}}^{(t)}$ defined above is given as the solution of the following:

$$p^{(t)} = \text{argmin}_{p \in \Delta} -s \langle p, G^{(t)} \rangle + D(p, p^{(t-1)}). \tag{15}$$

Here $D$ denotes the KL-divergence and we define $p^{(0)} = (1/K^2)_{1 \leq i,j \leq K} = \text{argmin}_{p \in \Delta} F(p)$. Since the optimization problem (15) is a convex optimization problem, by the first order optimality condition, we have

$$\langle \pi_0 - p^{(t)}, G^{(t)} \rangle \leq \frac{1}{s} \left\{ D(\pi_0, p^{(t-1)}) - D(\pi_0, p^{(t)}) - D(p^{(t)}, p^{(t-1)}) \right\}.$$

By summing the both sides and taking expectation, we have

$$T\overline{G}^{(T)}(\boldsymbol{\mathcal{P}}^{(0)}) - T\overline{G}^{(T)}(\boldsymbol{\mathcal{P}}_{\text{SelMix}}) \leq \frac{1}{s}\left\{ D(\pi_0, p^{(0)}) - D(\pi_0, p^{(T)}) - \sum_{t=1}^{T} D(p^{(t)}, p^{(t-1)}) \right\}$$

$$\leq \frac{1}{s}D(\pi_0, p^{(0)}).$$

Here the second inequality follows from the non-negativity of the KL-divergence. Since $p^{(0)} = \text{argmin}_p F(p)$, by the first-order optimality condition, we have $D(\pi_0, p^{(0)}) \leq F(\pi_0) - F(p^{(0)})$. Noting that $F(\pi_0) \leq 0$, we have the following

$$T\overline{G}^{(T)}(\boldsymbol{\mathcal{P}}^{(0)}) - T\overline{G}^{(T)}(\boldsymbol{\mathcal{P}}_{\text{SelMix}}) \leq \frac{-F(p^{(0)})}{s} = \frac{\log K^2}{s}.$$

This completes the proof. □

## C.5 A Variant of Theorem C.7

In the case when $\boldsymbol{\mathcal{P}}_{\text{SelMix}}^{(t)}$ is defined similarly to Hedge (Freund & Schapire, 1997), i.e., $\boldsymbol{\mathcal{P}}_{\text{SelMix}}^{(t)} = \text{softmax}((s\sum_{\tau=1}^{t-1} G_{ij}^{(\tau)})_{ij})$, then by the standard analysis, we can prove the following.

**Theorem C.8.** *We assume that $G_{ij}^{(t)}$ is normalized so that $|G_{ij}^{(t)}| \leq 1$. Then, with an appropriate choice of the parameter $s$, for any non-adaptive policy $\boldsymbol{\mathcal{P}}^{(0)} = (\mathcal{P}^{(0)})_{t=1}^{T}$, we have $\overline{G}^{(T)}(\boldsymbol{\mathcal{P}}_{\text{SelMix}}) + 2\sqrt{\log K}/\sqrt{T} \geq \overline{G}^{(T)}(\boldsymbol{\mathcal{P}}^{(0)})$.*

First we introduce the following lemma, which is due to (Freund & Schapire, 1997). Although, one can prove the following result by a standard argument, we provide a proof for the sake of completeness.

**Lemma C.9** (c.f. (Freund & Schapire, 1997)). *We assume that $G_{i,j}^{(t)} \in [0,1]$ for all $t$ and $1 \leq i,j \leq K$. For $(i,j) \in [K] \times [K]$, we define $\overline{S}_{i,j} = \sum_{t=1}^{T} \mathbb{E}\left[G_{i,j}^{(t)}\right]$. For a policy $\boldsymbol{\mathcal{P}} = (\mathcal{P}_t)_{t=1}^{T}$, we define $\overline{S}_{\boldsymbol{\mathcal{P}}} := \sum_{t=1}^{T} \mathbb{E}\left[G_{i_t,j_t}^{(t)}\right]$. Then, we have the following inequality:*

$$-2\log K + s\max_{(i,j)\in[K]\times[K]} \overline{S}_{i,j} \leq (\exp(s) - 1)\overline{S}_{\boldsymbol{\mathcal{P}}_{\text{SelMix}}}.$$

*Proof.* This lemma can be proved by a standard argument, but for the sake of completeness, we provide a proof. We put $\mathcal{A} = [K] \times [K]$, $a_t = (i^{(t)}, j^{(t)})$ and in the proof we simply denote $\boldsymbol{\mathcal{P}}_{\text{SelMix}}$ by $\boldsymbol{\mathcal{P}}$. For $a \in \mathcal{A}$ and $1 \leq t \leq T+1$, we define $w_{a,t}$ as follows. We define $w_{a,1} = 1/K^2$ for all $a \in \mathcal{A}$ and $w_{a,t+1} = w_{a,t}\exp(sG_a^{(t)})$. We also define $W_t = \sum_{a\in\mathcal{A}} w_{a,t}$. Then, the distribution $\mathcal{P}_t$ is given as the probability $(w_{a,t}/W_t)_{a\in\mathcal{A}}$ by definition. Noting that $\exp(sx) \leq 1 + (\exp(s) - 1)x$ for $x \in [0,1]$, we have the following inequality:

$$W_{t+1} = \sum_{a\in\mathcal{A}} w_{a,t+1} = \sum_{a\in\mathcal{A}} w_{a,t}\exp(sG_a^{(t)})$$

$$\leq \sum_{a\in\mathcal{A}} w_{a,t}(1 + \exp(s-1)G_a^{(t)}).$$

Thus, we have

$$W_{t+1} \leq \sum_{a\in\mathcal{A}} w_{a,t}(1 + \exp(s-1)G_a^{(t)})$$

$$= W_t\left(1 + (\exp(s) - 1)\mathbb{E}_{\mathcal{P}_t}\left[G_{a_t}^{(t)}\right]\right),$$

where $\mathbb{E}_{\mathcal{P}_t}[\cdot]$ denotes the expectation with respect to $a_t$. By repeatedly apply the inequality above, we obtain:

$$W_{T+1} \leq \prod_{t=1}^{T}\left(1 + (\exp(s) - 1)\mathbb{E}_{\mathcal{P}_t}\left[G_{a_t}^{(t)}\right]\right).$$

Let $a \in \mathcal{A}$ be any pair. By this inequality and $W_{T+1} \geq w_{a,T+1} = \frac{1}{K^2} \exp(s \sum_{t=1}^{T} G_a^{(t)})$, we have the following:

$$\frac{1}{K^2} \exp(s \sum_{t=1}^{T} G_a^{(t)}) \leq \prod_{t=1}^{T} \left( 1 + (\exp(s) - 1) \mathbb{E}_{\mathcal{P}_t} \left[ G_{a_t}^{(t)} \right] \right).$$

By taking log of both sides and $\log(1 + x) \leq x$, we have

$$-2 \log K + s \sum_{t=1}^{T} G_a^{(t)} \leq \sum_{t=1}^{T} \log \left( 1 + (\exp(s) - 1) \mathbb{E}_{\mathcal{P}_t} \left[ G_{a_t}^{(t)} \right] \right)$$

$$\leq (\exp(s) - 1) \sum_{t=1}^{T} \mathbb{E}_{\mathcal{P}_t} \left[ G_{a_t}^{(t)} \right].$$

By taking the expectation with respect to the randomness of $G_{i,j}^{(t)}$, we obtain the following:

$$-2 \log K + s \overline{S}_a \leq (\exp(s) - 1) \overline{S}_{\mathcal{P}}.$$

Since $a \in [K] \times [K]$ is arbitrary, we have the assertion of the lemma. $\qquad \square$

We can prove Theorem C.7 by Lemma C.9 as follows:

*Proof of Theorem C.7.* Let $(i^*, j^*)$ be the best fixed Mixup pair hindsight, i.e., $(i^*, j^*) = \operatorname{argmax}_{(i,j) \in [K] \times [K]} \overline{S}_{i,j}$. Since any non-adaptive (or stationary) policy is no better than $\delta_{(i^*,j^*)}$, to prove the theorem, it is enough to prove the following:

$$\overline{S}_{i^*,j^*} \leq \overline{S}_{\mathcal{P}} + 2\sqrt{T \log K}. \tag{16}$$

Here in this proof, we simply denote $\mathcal{P}_{\mathrm{SelMix}}$ by $\mathcal{P}$. To prove (16), we define the pseudo regret $R_T$ by $R_T = \overline{S}_{i^*,j^*} - \overline{S}_{\mathcal{P}}$. Then by Lemma C.9, we have

$$R_T \leq \frac{(\exp(s) - 1 - s)\overline{S}_{i^*,j^*} + 2 \log K}{\exp(s) - 1}.$$

We put $s = \log(1 + \alpha)$ with $\alpha > 0$. Then we have

$$R_T \leq \frac{(\alpha - \log(1 + \alpha))\overline{S}_{i^*,j^*} + 2 \log K}{\alpha}.$$

We note that the following inequality holds for $\alpha > 0$:

$$\frac{\alpha - \log(1 + \alpha)}{\alpha} \leq \frac{1}{2}\alpha$$

Then it follows that:

$$R_T \leq \frac{1}{2}\alpha \overline{S}_{i^*,j^*} + \frac{2 \log K}{\alpha} \leq \frac{1}{2}\alpha T + \frac{2 \log K}{\alpha}.$$

Here the second inequality follows from $\overline{S}_{i^*,j^*} \leq T$. We take $\alpha = 2\sqrt{\frac{\log K}{T}}$. Then we have $R_T \leq 2\sqrt{T \log K}$. Thus, we have the assertion of the theorem. $\qquad \square$

## D  UNCONSTRAINED DERIVATIVES OF METRIC

For any general metric $\psi(C[h])$ the derivative w.r.t the unconstrained confusion matrix $\tilde{C}[h]$ is expressible purely in terms of the entries of the confusion matrix. This is because $C[h] = \operatorname{softmax}(\tilde{C}[h])$ The derivative using the chain rule is expressed as follows,

$$\frac{\partial \psi(C[h])}{\partial \tilde{C}_{ij}[h]} = \sum_{k,l} \frac{\partial \psi(C[h])}{\partial C_{kl}[h]} \cdot \frac{\partial C_{kl}[h]}{\partial \tilde{C}_{ij}[h]} \tag{17}$$

We observe that in Eq. 17 the partial derivative $\frac{\partial \psi(C[h])}{\partial C_{kl}[h]}$ is purely a function of entries of $C[h]$ since $\psi(C[h])$ itself is a function of entries of $C[h]$. The second term is the partial derivative of our confusion matrix w.r.t the unconstrained confusion matrix. Since $C$ and $\tilde{C}$ are related by the following relation $C_{ij}[h] = \text{softmax}(\tilde{C}_i[h])_j$. By virtue of the aforementioned map $\frac{\partial C_{kl}[h]}{\partial \tilde{C}_{ij}[h]}$ also happens to be expressible in terms of $C[h]$:

$$\frac{\partial C_{kl}[h]}{\partial \tilde{C}_{ij}[h]} = \begin{cases} 0, & k \neq i \\ -\frac{C_{il}[h] \cdot C_{ij}[h]}{\pi_i^{\text{val}}}, & i = k, l \neq j \\ C_{ij}[h] - \frac{C_{ij}^2[h]}{(\pi_i^{\text{val}})^2}, & i = k, l = j \end{cases} \tag{18}$$

Let us consider the metric mean recall $\psi^{\text{AM}}(C[h]) = \frac{1}{K} \sum_i \frac{C_{ii}[h]}{\sum_j C_{ij}[h]}$. The derivative of $\psi^{\text{AM}}(C[h])$ w.r.t the unconstrained confusion matrix $\tilde{C}$ can be expressed in terms of the entries of the confusion matrix. This is a useful property of this partial derivative since we need not infer the inverse map from $C \to \tilde{C}$ inorder to evaluate the partial derivative in terms of $\tilde{C}$. It can be expressed follows:

$$\frac{\partial \psi^{\text{AM}}(C[h])}{\partial \tilde{C}_{ij}[h]} = \begin{cases} -\frac{C_{ij}[h] \cdot C_{ii}[h]}{K(\pi_i^{\text{val}})^2}, & m \neq n \\ \frac{C_{ii}[h]}{K \cdot \pi_i^{\text{val}}} - \frac{C_{ii}^2[h]}{K \cdot (\pi_i^{\text{val}})^2}, & m = n \end{cases} \tag{19}$$

Hence we can conclude that for a metric defined as a function of the entries of the confusion matrix, the derivative w.r.t the unconstrained confusion matrix ($\tilde{C}$) is easily expressible using the entries of the confusion matrix ($C$).

## E  COMPARISON OF SELMIX WITH OTHER FRAMEWORKS

In our work, we optimize the non-decomposable objective function by using Mixup (Zhang et al., 2018). In recent works, Mixup training has been shown to be effective specifically for real-world scenarios where the long-tailed imbalance is present in the dataset (Zhong et al., 2021; Fan et al., 2022). Further, mixup has been demonstrated to have a data-dependent regularization effect (Zhang et al., 2021). Hence, this provides us the motivation to consider optimization of the non-decomposable objectives which are important for long-tailed imbalanced datasets, in terms of directions induced by Mixups.However, this mixup-induced data-dependent regularization is not present for works (Narasimhan & Menon, 2021; Rangwani et al., 2022), which use consistent loss functions without mixup. Hence, this explains the superior generalization demonstrated by SelMix (mixup based) on non-decomposable objective optimization for long-tailed datasets.

## F  UPDATING THE LAGRANGE MULTIPLIERS

### F.1  MIN. RECALL AND MIN OF HEAD AND TAIL RECALL

Consider the objective of optimizing the worst-case(Min.) recall, $\psi^{\text{MR}}(C[h]) = \min_{\boldsymbol{\lambda} \in \Delta_{K-1}} \sum_{i \in [K]} \lambda_i \text{Rec}_i[h] = \min_{\boldsymbol{\lambda} \in \Delta_{K-1}} \sum_{i \in [K]} \lambda_i \frac{C_{ii}[h]}{\sum_{j \in [K]} C_{ij}[h]}$, as in Table G.3. the Lagrange multipliers are sampled from a $K - 1$ dimensional simplex and $\lambda_i = 1$ if recall of $i^{\text{th}}$ class is the lowest and the remaining lagrange multipliers are zero. Hence, a good approximation of the lagrange multipliers at a given time step $t$ can be expressed as:

$$\lambda_i^{(t)} = \frac{e^{-\omega \text{Rec}_i^{(t)}[h]}}{\sum_{j \in [K]} e^{-\omega \text{Rec}_j^{(t)}[h]}} \tag{20}$$

This has some nice properties such as the Lagrange multipliers being a soft and momentum-free approximation of their hard counterpart. This enables SelMix to compute a sampling distribution $\mathcal{P}_{\text{SelMix}}^{(t)}$ that neither over-corrects nor undercorrects based on the feedback from the validation set. For sufficiently high $\omega$ this approximates the objective to the min recall.

### F.2 MEAN RECALL UNDER COVERAGE CONSTRAINTS

For the objective where we wish to optimize for the mean recall, subject to the constraint that all the classes have a coverage above $\frac{\alpha}{K}$, where if $\alpha = 1$ is the ideal case for a balanced validation set. We shall relax this constraint and set $\alpha = 0.95$, $\psi_{\text{cons.}}^{\text{AM}}(C[h]) = \min_{\boldsymbol{\lambda} \in \mathbb{R}_+^K} \frac{1}{K} \sum_{i \in [K]} \text{Rec}_i[h] + \sum_{j \in [K]} \lambda_j \left( \text{Cov}_j[h] - \frac{\alpha}{K} \right) = \min_{\boldsymbol{\lambda} \in \mathbb{R}_+^K} \frac{1}{K} \sum_{i \in [K]} \frac{C_{ii}[h]}{\sum_{j \in [K]} C_{ij}[h]} + \sum_{j \in [K]} \lambda_j \left( \sum_{i \in [K]} C_{ij}[h] - \frac{\alpha}{K} \right)$.
For practical purposes, we look at a related constrained optimization problem,

$$\psi_{\text{cons.}}^{\text{AM}}(C[h]) = \min_{\boldsymbol{\lambda} \in \mathbb{R}_+^K} \frac{1}{\lambda_{max} + 1} \left( \frac{1}{K} \sum_{i \in [K]} \text{Rec}_i[h] + \sum_{j \in [K]} \lambda_j \left( \text{Cov}_j[h] - \frac{\alpha}{K} \right) \right)$$

Such that if $\left( \text{Cov}_i[h] - \frac{\alpha}{K} \right) < 0$, then $\lambda_i$ increases, and vice-versa for the converse case. Also, if $\exists i$ s.t. $\left( \text{Cov}_i[h] - \frac{\alpha}{K} \right) < 0$, then this implies that $\frac{1}{\lambda_{\max}+1} \to 0^+$ and $\frac{\lambda_{\max}}{\lambda_{\max}+1} \to 1^-$, which forces $h$ to satisfy the constraint $\left( \text{Cov}_i[h] - \frac{\alpha}{K} \right) > 0$. Based on this, a momentum free formulation for updating the lagrangian multipliers is as follows:

$$\lambda_i = \max \left( 0, \Lambda_{\max} \left( 1 - e^{\frac{\text{Cov}_i[h] - \frac{\alpha}{K}}{\tau}} \right) \right)$$

Here, $\lambda_{max}$ is the maximum value that the Lagrange multiplier can take. A large value of $\lambda_{max}$ forces the model to focus more on the coverage constraints that to be biased towards mean recall optimization. $\tau$ is a hyperparameter that is usually kept small, say 0.01 or so, which acts as sort of a tolerance factor to keep the constraint violation in check.

## G EXPERIMENTAL DETAILS

The baselines (Table 2) are evaluated with the SotA base pre-training method of FixMatch + LA using DASO codebase (Oh et al., 2022), whereas CSST (Rangwani et al., 2022) is done through their official codebase .

### G.1 HYPERPARAMETER TABLE

The detailed values of all hyperparameters specific to each dataset has been mentioned in Table G.1.

Table G.1: Table of Hyperparameters for Semi-Supervised datasets.

| Parameter | CIFAR-10 (All distributions) | CIFAR-100 ($\rho_l = 10, \rho_u = 10$) | STL-10 | Imagenet-100($\rho_l = \rho_u = 10$) |
|---|---|---|---|---|
| Gain scaling ($s$) | 10.0 | 10.0 | 2.0 | 10.0 |
| $\omega_{\text{Min. Rec}}$ | 40 | 20 | 20 | 20 |
| $\lambda_{\max}$ | 100 | 100 | 100 | 100 |
| $\tau$ | 0.01 | 0.01 | 0.01 | 0.01 |
| $\alpha$ | 0.95 | 0.95 | 0.95 | 0.95 |
| Batch Size | 64 | 64 | 64 | 64 |
| Learning Rate($f$) | 3e-4 | 3e-4 | 3e-4 | 0.1 |
| Learning Rate($g$) | 3e-5 | 3e-5 | 3e-5 | 0.01 |
| Optimizer | SGD | SGD | SGD | SGD |
| Scheduler | Cosine | Cosine | Cosine | Cosine |
| Total SGD Steps | 10k | 10k | 10k | 10k |
| Resolution | 32 X 32 | 32 X 32 | 32 X 32 | 224 X 224 |
| Arch. | WRN-28-2 | WRN-28-2 | WRN-28-2 | WRN-28-2 |

Table G.2: Table of Hyperparameters for Supervised Datasets.

| Parameter | CIFAR-10 ($\rho = 100$) | CIFAR-100 ($\rho = 100$) | Imagenet-1k LT |
|---|---|---|---|
| Gain scaling ($s$) | 10.0 | 10.0 | 10.0 |
| $\omega_{\text{Min. Rec}}$ | 50 | 25 | 100 |
| $\lambda_{\max}$ | 100 | 100 | 100 |
| $\tau$ | 0.01 | 0.01 | 0.001 |
| $\alpha$ | 0.95 | 0.95 | 0.95 |
| Batch Size | 128 | 128 | 256 |
| Learning Rate($f$) | 3e-3 | 3e-3 | 0.1 |
| Learning Rate($g$) | 3e-4 | 3e-4 | 0.01 |
| Optimizer | SGD | SGD | SGD |
| Scheduler | Cosine | Cosine | Cosine |
| Total SGD Steps | 2k | 2k | 2.5k |
| Resolution | 32 X 32 | 32 X 32 | 224 X 224 |
| Arch. | ResNet-32 | ResNet-32 | ResNet-50 |

## G.2 EXPERIMENTAL DETAILS FOR SUPERVISED LEARNING

We show our results on 3 datasets: CIFAR-10 LT ($\rho = 100$), CIFAR-100 LT ($\rho = 100$) (Krizhevsky & Hinton, 2009) and Imagenet-1k (Russakovsky et al., 2015) LT. For our pre-trained model, we use the model trained by stage-1 of MiSLAS(Zhong et al., 2021), which uses a mixup-based pre-training as it improves calibration. For CIFAR-10,100 LT ($\rho = 100$) we use ResNet-32 while for Imagenet-1k LT, we use ResNet-50. The detailed list of hyperparameters have been provided in Tab. G.2. Unlike semi-supervised fine-tuning, we do not require to refresh the pseudo-labels for the unlabelled samples since we already have the true labels. The backbone is trained at a learning rate $\frac{1}{10}^{th}$ of the linear classifier learning rate. The batch norm is frozen across all the layers. The detailed algorithm can be found in Alg. 2 and is very similar to its semi-supervised variant. We report the performance obtained at the end of fine-tuning.

---

**Algorithm 2** Training through SelMix .

---

**Input:** Data $(D, D^{\text{val}})$, iterations $T$, classifier $h^{(0)}$, metric function $\psi$
**for** $t = 1$ **to** $T$ **do**
   $h^{(t)} = h^{(t-1)}$, $C^{(t)} = \mathbb{E}_{(x,y)\sim D^{\text{val}}}[C[h^{(t)}]]$
   $V_{ij}^{(t)} = -\eta \frac{\partial \mathcal{L}_{ij}^{\text{mix}}}{\partial W} \quad \forall\, i,j \qquad$ (4)
   $G_{ij}^{(t)} = \sum_{k,l} \frac{\partial \psi(C^{(t)})}{\partial \tilde{C}_{kl}} (V_{ij}^{(t)})_l^\top \cdot z_k \quad \forall\, i,j$
   $\mathcal{P}_{\text{SelMix}}^{(t)} = \text{softmax}(s\mathbf{G}^{(t)})$ // *Compute Sampling distribution and update pseudo-label*
   **for** $n$ SGD steps **do**
      $Y_1, Y_2 \sim \mathcal{P}_{\text{SelMix}}^{(t)}$
      $X_1 \sim \mathcal{U}(D_{Y_1})$, $X_2 \sim \mathcal{U}(D_{Y_2})$    // *sample batches of data*
      $h^{(t)} := \text{SGD-Update}(h^{(t)}, \mathcal{L}_{\text{mixup}}, (X_1, Y_1, X_2))$
   **end for**
**end for**
**Output:** $h^{(T)}$

---

## H RESULTS FOR CASE WITH UNKNOWN LABELED DISTRIBUTION

In this section, we provide a full-scale comparison of all the methods of the case when the labeled distribution does not match the unlabeled data distribution, simulating the scenario when label distribution is unknown. The main paper presents the summary plots for these results in Fig. 3. The Table H.1, we present results for the case when for CIFAR-10 once the unlabeled data follow an inverse label distribution i.e. ($\rho_l = 100, \rho_u = \frac{1}{100}$) and the case when the unlabeled data is distributed

Table G.3: The Expression of Non-Decomposable Objectives we consider in our paper.

| Metric | Definition |
|---|---|
| Mean Recall ($\psi^{AM}$) | $\frac{1}{K}\sum_{i\in[K]}\frac{C_{ii}[h]}{\sum_{j\in[K]}C_{ij}[h]}$ |
| G-mean ($\psi^{GM}$) | $\left(\prod_{i\in[K]}\frac{C_{ii}[h]}{\sum_{j\in[K]}C_{ij}[h]}\right)^{\frac{1}{k}}$ |
| H-mean ($\psi^{HM}$) | $K\left(\sum_{i\in[K]}\frac{\sum_{j\in[K]}C_{ij}[h]}{C_{ii}[h]}\right)^{-1}$ |
| Min. Recall ($\psi^{MR}$) | $\min_{\boldsymbol{\lambda}\in\Delta_{K-1}}\sum_{i\in[K]}\lambda_i\frac{C_{ii}[h]}{\sum_{j\in[K]}C_{ij}[h]}$ |
| Min of Head and Tail class recall ($\psi^{MR}_{HT}$) | $\min_{(\lambda_{\mathcal{H}},\lambda_{\mathcal{T}})\in\Delta_1}\frac{\lambda_{\mathcal{H}}}{|\mathcal{H}|}\sum_{i\in\mathcal{H}}\frac{C_{ii}[h]}{\sum_{j\in[K]}C_{ij}[h]}$ $+\frac{\lambda_{\mathcal{T}}}{|\mathcal{T}|}\sum_{i\in\mathcal{T}}\frac{C_{ii}[h]}{\sum_{j\in[K]}C_{ij}[h]}$ |
| Mean Recall s.t. per class coverage $\geq\frac{\alpha}{K}$ ($\psi^{AM}_{\text{cons.}}$) | $\min_{\boldsymbol{\lambda}\in\mathbb{R}^K_+}\frac{1}{K}\sum_{i\in[K]}\frac{C_{ii}[h]}{\sum_{j\in[K]}C_{ij}[h]}+\sum_{j\in[K]}\lambda_j\left(\sum_{i\in[K]}C_{ij}[h]-\frac{\alpha}{K}\right)$ |
| Mean Recall s.t. minimum of head and tail class coverage $\geq\frac{\alpha}{K}$ ($\psi^{AM}_{\text{cons.(HT)}}$) | $\min_{(\lambda_{\mathcal{H}},\lambda_{\mathcal{T}})\in\mathbb{R}^2_{\geq 0}}\frac{1}{K}\sum_{i\in[K]}\frac{C_{ii}[h]}{\sum_{j\in[K]}C_{ij}[h]}+\lambda_{\mathcal{H}}\left(\sum_{i\in[K],j\in\mathcal{H}}\frac{C_{ij}[h]}{|\mathcal{H}|}-\frac{0.95}{K}\right)$ $+\lambda_{\mathcal{T}}\left(\sum_{i\in[K],j\in\mathcal{T}}\frac{C_{ij}[h]}{|\mathcal{T}|}-\frac{0.95}{K}\right)$ |
| H-mean s.t. per class coverage $\geq\frac{\alpha}{K}$ ($\psi^{HM}_{\text{cons.}}$) | $\min_{\boldsymbol{\lambda}\in\mathbb{R}^K_+}K\left(\sum_{i\in[K]}\frac{\sum_{j\in[K]}C_{ij}[h]}{C_{ii}[h]}\right)^{-1}+\sum_{j\in[K]}\lambda_j\left(\sum_{i\in[K]}C_{ij}[h]-\frac{\alpha}{K}\right)$ |
| H-mean s.t. minimum of head and tail class coverage $\geq\frac{\alpha}{K}$ ($\psi^{HM}_{\text{cons.(HT)}}$) | $\min_{(\lambda_{\mathcal{H}},\lambda_{\mathcal{T}})\in\mathbb{R}^2_{\geq 0}}K\left(\sum_{i\in[K]}\frac{\sum_{j\in[K]}C_{ij}[h]}{C_{ii}[h]}\right)^{-1}+\lambda_{\mathcal{H}}\left(\sum_{i\in[K],j\in\mathcal{H}}\frac{C_{ij}[h]}{|\mathcal{H}|}-\frac{0.95}{K}\right)$ $+\lambda_{\mathcal{T}}\left(\sum_{i\in[K],j\in\mathcal{T}}\frac{C_{ij}[h]}{|\mathcal{T}|}-\frac{0.95}{K}\right)$ |

uniformly across all classes ($\rho_l=100,\rho_u=1$). In both cases, we find that SelMix can produce significant improvement across metrics. Further, we also compare our method in the practical setup where unlabeled data distribution is unknown. This situation is perfectly emulated by the STL-10 dataset, which also contains an unlabeled set of 100k images. Table H.2 presents results for different approaches on the STL-10 case. We observe that SelMix produces superior results compared to baselines and is robust to the mismatch in distribution between labeled and unlabeled data.

Table H.1: Comparison on metric objectives for CIFAR-10 LT under $\rho_l\neq\rho_u$ assumption. Our experiments involve $\rho_u=100,\rho_l=1$ (uniform) and $\rho_u=100,\rho_l=\frac{1}{100}$ (inverted). SelMix achieves significant gains over other SSL-LT methods across all the metrics.

| | CIFAR-10 ($\rho_l=100,\rho_u=\frac{1}{100},N_1=1500,M_1=30$) | | | | | CIFAR-10 ($\rho_l=100,\rho_u=1,N_1=1500,M_1=3000$) | | | | |
|---|---|---|---|---|---|---|---|---|---|---|
| | Mean Rec. | Min Rec. | HM | GM | Mean Rec./Min Cov. | Mean Rec. | Min Rec. | HM | GM | Mean Rec./Min Cov. |
| FixMatch | $71.3_{\pm1.1}$ | $28.5_{\pm2.6}$ | $61.3_{\pm2.7}$ | $67.1_{\pm1.7}$ | $71.3_{\pm1.1}/0.030_{\pm2e\text{-}3}$ | $82.8_{\pm1.3}$ | $59.1_{\pm5.8}$ | $80.6_{\pm2.1}$ | $82.3_{\pm1.5}$ | $82.8_{\pm1.3}/0.059_{\pm6e\text{-}3}$ |
| DARP | $79.7_{\pm0.8}$ | $60.7_{\pm2.4}$ | $78.1_{\pm0.9}$ | $78.9_{\pm0.9}$ | $79.7_{\pm0.8}/0.065_{\pm2e\text{-}3}$ | $84.8_{\pm0.7}$ | $66.9_{\pm3.1}$ | $83.5_{\pm0.8}$ | $85.2_{\pm0.7}$ | $84.8_{\pm0.7}/0.067_{\pm3e\text{-}3}$ |
| CReST | $71.3_{\pm0.9}$ | $40.3_{\pm3.0}$ | $65.8_{\pm1.5}$ | $68.6_{\pm1.2}$ | $71.3_{\pm0.9}/0.040_{\pm5e\text{-}3}$ | $85.7_{\pm0.3}$ | $68.7_{\pm1.7}$ | $84.6_{\pm0.14}$ | $85.1_{\pm0.1}$ | $85.7_{\pm0.3}/0.075_{\pm7e\text{-}4}$ |
| CReST+ | $72.8_{\pm0.8}$ | $45.2_{\pm2.5}$ | $68.4_{\pm1.3}$ | $70.6_{\pm1.1}$ | $72.8_{\pm0.8}/0.047_{\pm3e\text{-}3}$ | $86.4_{\pm0.2}$ | $71.7_{\pm1.9}$ | $85.6_{\pm0.2}$ | $86.1_{\pm0.1}$ | $86.4_{\pm0.2}/0.078_{\pm1e\text{-}3}$ |
| DASO | $79.2_{\pm0.2}$ | $64.6_{\pm1.9}$ | $78.1_{\pm0.1}$ | $78.6_{\pm0.8}$ | $79.2_{\pm0.2}/0.072_{\pm3e\text{-}3}$ | $88.6_{\pm0.4}$ | $78.2_{\pm1.6}$ | $88.4_{\pm0.5}$ | $88.5_{\pm0.4}$ | $88.6_{\pm0.4}/0.089_{\pm1e\text{-}3}$ |
| ABC | $80.8_{\pm0.4}$ | $65.1_{\pm0.8}$ | $79.6_{\pm0.3}$ | $80.7_{\pm0.6}$ | $80.8_{\pm0.4}/0.073_{\pm5e\text{-}3}$ | $88.6_{\pm0.4}$ | $74.8_{\pm2.9}$ | $88.2_{\pm0.7}$ | $88.6_{\pm0.3}$ | $88.6_{\pm0.4}/0.086_{\pm4e\text{-}3}$ |
| CoSSL | $78.6_{\pm1.0}$ | $66.3_{\pm2.9}$ | $77.2_{\pm1.2}$ | $77.8_{\pm1.1}$ | $78.6_{\pm1.0}/0.070_{\pm2e\text{-}3}$ | $88.7_{\pm0.9}$ | $76.1_{\pm2.9}$ | $88.2_{\pm1.1}$ | $88.5_{\pm1.0}$ | $88.7_{\pm0.9}/0.084_{\pm8e\text{-}3}$ |
| CSST | $77.5_{\pm1.5}$ | $72.1_{\pm0.2}$ | $76.5_{\pm4.9}$ | $76.8_{\pm5.2}$ | $77.5_{\pm1.5}/0.091_{\pm3e\text{-}3}$ | $87.6_{\pm0.7}$ | $78.1_{\pm0.3}$ | $86.1_{\pm0.7}$ | $87.1_{\pm0.2}$ | $87.6_{\pm0.7}/0.091_{\pm1e\text{-}3}$ |
| FixMatch(LA) | $75.5_{\pm1.5}$ | $45.1_{\pm4.4}$ | $71.1_{\pm2.5}$ | $73.3_{\pm1.9}$ | $75.5_{\pm1.5}/0.046_{\pm4e\text{-}3}$ | $90.1_{\pm0.4}$ | $75.8_{\pm2.1}$ | $89.5_{\pm0.7}$ | $89.7_{\pm0.5}$ | $90.1_{\pm0.4}/0.083_{\pm1e\text{-}3}$ |
| **w/SelMix (Ours)** | $\mathbf{81.3}_{\pm0.5}$ | $\mathbf{74.3}_{\pm1.2}$ | $\mathbf{81.0}_{\pm0.8}$ | $\mathbf{80.9}_{\pm0.5}$ | $\mathbf{81.7}_{\pm0.8}/\mathbf{0.091}_{\pm3e\text{-}3}$ | $\mathbf{91.4}_{\pm1.2}$ | $\mathbf{84.7}_{\pm0.7}$ | $\mathbf{91.3}_{\pm1.1}$ | $\mathbf{91.3}_{\pm1.2}$ | $\mathbf{91.4}_{\pm1.2}/\mathbf{0.096}_{\pm1e\text{-}3}$ |

# I  OPTIMIZATION OF H-MEAN WITH COVERAGE CONSTRAINTS

We consider the objective of optimizing H-mean subject to the constraint that all classes must have a coverage $\geq\frac{\alpha}{K}$. For CIFAR-10, when the unlabeled data distribution matches the labeled data distribution, uniform or inverted, SelMix is able to satisfy the coverage constraints. A similar observation could be made for CIFAR-100, where the constraint is to have the minimum head and tail class coverage above $\frac{0.95}{K}$. For STL-10, SelMix fails to satisfy the constraint because the validation dataset is minimal (500 samples compared to 5000 in CIFAR). We want to convey here that as CSST is only able to optimize for linear metrics like min. recall its performance is inferior on complex objectives like optimizing H-mean with constraints. This shows the superiority of the proposed SelMix framework.

Table H.2: Comparison across methods when label distribution $\rho_u$ is unknown. We use the STL-10 dataset for comparison in such a case.

| | STL-10 | | | $(\rho_l = 10, \rho_u = \text{NA}, N_1 = 450, \sum_i M_i = 100\text{k})$ | | |
|---|---|---|---|---|---|---|
| | Mean Rec. | Min Rec. | HM | GM | HM/Min Cov. | Mean Rec./Min Cov. |
| FixMatch | $72.7_{\pm 0.7}$ | $43.2_{\pm 7.1}$ | $67.7_{\pm 1.5}$ | $71.6_{\pm 1.3}$ | $67.7_{\pm 1.5}$ / $0.048_{\pm 1\text{e-}2}$ | $72.7_{\pm 0.7}$ / $0.048_{\pm 1\text{e-}2}$ |
| DARP | $76.5_{\pm 0.3}$ | $54.7_{\pm 1.9}$ | $74.0_{\pm 0.5}$ | $75.3_{\pm 0.4}$ | $74.0_{\pm 0.5}$ / $0.058_{\pm 2\text{e-}3}$ | $76.5_{\pm 0.3}$ / $0.058_{\pm 2\text{e-}3}$ |
| CReST | $70.1_{\pm 0.3}$ | $48.2_{\pm 2.2}$ | $67.1_{\pm 1.1}$ | $67.8_{\pm 1.1}$ | $67.1_{\pm 1.1}$ / $0.066_{\pm 2\text{e-}3}$ | $70.1_{\pm 0.3}$ / $0.066_{\pm 2\text{e-}3}$ |
| DASO | $78.1_{\pm 0.5}$ | $55.8_{\pm 3.7}$ | $76.6_{\pm 1.1}$ | $77.2_{\pm 0.2}$ | $76.6_{\pm 1.1}$ / $0.083_{\pm 3\text{e-}3}$ | $78.1_{\pm 0.5}$ / $0.083_{\pm 3\text{e-}3}$ |
| ABC | $77.5_{\pm 0.4}$ | $55.4_{\pm 6.7}$ | $74.7_{\pm 1.5}$ | $76.3_{\pm 0.9}$ | $74.7_{\pm 1.5}$ / $0.079_{\pm 7\text{e-}3}$ | $77.5_{\pm 0.4}$ / $0.079_{\pm 7\text{e-}3}$ |
| CSST | $79.2_{\pm 1.5}$ | $50.8_{\pm 2.9}$ | $78.3_{\pm 2.6}$ | $78.9_{\pm 2.1}$ | $78.3_{\pm 2.6}$ / $0.081_{\pm 6\text{e-}3}$ | $79.2_{\pm 1.5}$ / $0.081_{\pm 6\text{e-}3}$ |
| FixMatch(LA) | $78.9_{\pm 0.4}$ | $56.4_{\pm 1.9}$ | $76.5_{\pm 1.1}$ | $77.8_{\pm 0.8}$ | $76.5_{\pm 1.1}$ / $0.066_{\pm 5\text{e-}3}$ | $78.9_{\pm 0.4}$ / $0.066_{\pm 5\text{e-}3}$ |
| **w/SelMix (Ours)** | $\mathbf{80.9}_{\pm 0.5}$ | $\mathbf{68.5}_{\pm 1.8}$ | $\mathbf{79.1}_{\pm 1.2}$ | $\mathbf{80.1}_{\pm 0.4}$ | $\mathbf{79.1}_{\pm 1.2}$ / $\mathbf{0.088}_{\pm 1\text{e-}3}$ | $\mathbf{80.9}_{\pm 0.1}$ / $\mathbf{0.088}_{\pm 1\text{e-}3}$ |

Table I.1: Comparison of methods for optimization of H-mean with coverage constraints.

| | CIFAR-10 $\rho_l = 100, \rho_u = \frac{1}{100}$ $N_1 = 1500, M_1 = 30$ | | CIFAR-10 $\rho_l = \rho_u = 100$ $N_1 = 1500, M_1 = 3000$ | | CIFAR-10 $\rho_l = 100, \rho_u = 1$ $N_1 = 1500, M_1 = 3000$ | | CIFAR-100 $\rho_l = \rho_u = 10$ $N_1 = 150, M_1 = 300$ | | STL-10 $\rho_l = 10, \rho_u = \text{NA}$ $N_1 = 450, \sum_i M_i = 100\text{k}$ | |
|---|---|---|---|---|---|---|---|---|---|---|
| | HM | Min Cov. | HM | Min Cov. | HM | Min Cov. | HM | Min H-T Cov. | HM | Min Cov. |
| DARP | $78.1_{\pm 0.9}$ | $0.065_{\pm 3\text{e-}3}$ | $81.9_{\pm 0.5}$ | $0.070_{\pm 3\text{e-}3}$ | $83.5_{\pm 0.8}$ | $0.067_{\pm 3\text{e-}3}$ | $48.7_{\pm 1.3}$ | $0.0040_{\pm 2\text{e-}3}$ | $74.0_{\pm 0.5}$ | $0.058_{\pm 2\text{e-}3}$ |
| CReST | $65.8_{\pm 1.5}$ | $0.040_{\pm 5\text{e-}3}$ | $81.0_{\pm 0.7}$ | $0.073_{\pm 5\text{e-}3}$ | $84.6_{\pm 0.2}$ | $0.075_{\pm 7\text{e-}4}$ | $48.3_{\pm 0.2}$ | $0.0083_{\pm 2\text{e-}4}$ | $67.1_{\pm 1.1}$ | $0.066_{\pm 2\text{e-}3}$ |
| DASO | $78.1_{\pm 0.1}$ | $0.072_{\pm 3\text{e-}3}$ | $83.5_{\pm 0.3}$ | $0.083_{\pm 1\text{e-}3}$ | $88.4_{\pm 0.5}$ | $0.089_{\pm 1\text{e-}3}$ | $49.1_{\pm 0.7}$ | $0.0063_{\pm 3\text{e-}4}$ | $76.6_{\pm 1.1}$ | $0.083_{\pm 3\text{e-}3}$ |
| ABC | $\underline{79.6}_{\pm 0.3}$ | $0.073_{\pm 5\text{e-}3}$ | $\underline{84.6}_{\pm 0.5}$ | $0.086_{\pm 3\text{e-}3}$ | $88.2_{\pm 0.7}$ | $0.086_{\pm 1\text{e-}3}$ | $\underline{50.1}_{\pm 1.2}$ | $0.0089_{\pm 2\text{e-}4}$ | $74.7_{\pm 1.5}$ | $0.079_{\pm 7\text{e-}3}$ |
| CSST | $76.5_{\pm 4.9}$ | $\underline{0.081}_{\pm 6\text{e-}3}$ | $76.9_{\pm 0.2}$ | $\underline{0.093}_{\pm 3\text{e-}4}$ | $86.7_{\pm 0.7}$ | $\underline{0.092}_{\pm 1\text{e-}3}$ | $47.7_{\pm 0.8}$ | $\underline{0.0098}_{\pm 2\text{e-}4}$ | $\underline{78.3}_{\pm 2.6}$ | $0.081_{\pm 6\text{e-}3}$ |
| FixMatch (LA) | $78.3_{\pm 0.8}$ | $0.064_{\pm 1\text{e-}3}$ | $76.7_{\pm 0.1}$ | $0.056_{\pm 3\text{e-}3}$ | $\underline{89.3}_{\pm 0.2}$ | $0.086_{\pm 1\text{e-}3}$ | $45.5_{\pm 2.1}$ | $0.0053_{\pm 1\text{e-}4}$ | $74.6_{\pm 1.7}$ | $0.066_{\pm 5\text{e-}3}$ |
| **w/SelMix (Ours)** | $\mathbf{81.0}_{\pm 0.8}$ | $\mathbf{0.095}_{\pm 1\text{e-}3}$ | $\mathbf{85.1}_{\pm 0.1}$ | $\mathbf{0.095}_{\pm 1\text{e-}3}$ | $\mathbf{91.3}_{\pm 0.7}$ | $\mathbf{0.096}_{\pm 1\text{e-}3}$ | $\mathbf{53.8}_{\pm 0.5}$ | $\mathbf{0.0098}_{\pm 1\text{e-}4}$ | $\mathbf{79.1}_{\pm 1.2}$ | $\mathbf{0.088}_{\pm 1\text{e-}3}$ |

(a) Initial Stage ($t = 0$ SGD steps)     (b) Intermediate Stage ($t = 5k$ SGD steps)     (c) Final Stage ($t = 10k$ SGD steps)

Figure J.1: Evolution of gain matrix for mean recall optimized run for CIFAR-10 LT ($\rho_l = \rho_u$).

## J  EVOLUTION OF GAIN MATRIX WITH TRAINING

From the above collection of gain matrices, which are taken from different time steps of the training phase, we observe that $(|\max(\mathbf{G}^{(t)})|)$ of the gain matrix decreases with increase in SGD steps $t$, and settles on a negligible value by the time training is finished. This could be attributed to the fact that as the training progresses, the marginal improvement of the gain matrix decreases.

Another phenomenon we observe is that initially, during training, only a few mixups (particularly tail class ones) have a disproportionate amount of gain associated with them. A downstream consequence of this is that the sampling function $\mathcal{P}_{\text{SelMix}}$ prefers only a few $(i, j)$ mixups. Whereas, as the training continues, it becomes more exploratory rather than greedily exploiting the mixups that give the maximum gain at a particular timestep.

## K  ANALYSIS

We use a subset of Min. Recall, Mean Recall and Mean Recall with constraints for analysis.

**a) Scalability and Computation.** To demonstrate scalability of our method we show results on ImageNet100-LT for semi and ImageNet-LT for fully supervised settings. Similar to other datasets,

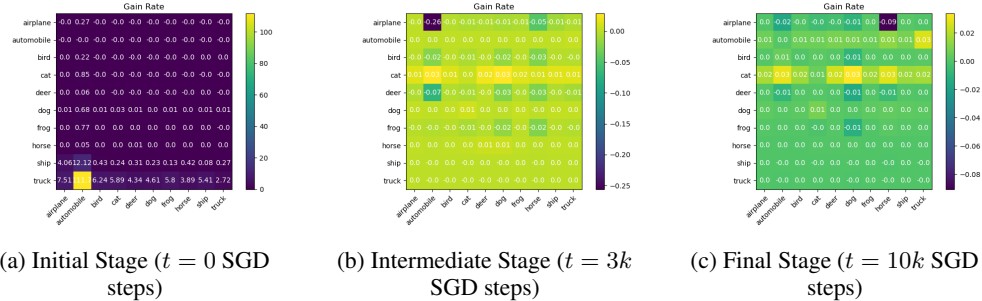

(a) Initial Stage ($t = 0$ SGD steps)

(b) Intermediate Stage ($t = 3k$ SGD steps)

(c) Final Stage ($t = 10k$ SGD steps)

Figure J.2: Evolution of gain matrix for min. recall optimized run for CIFAR-10 LT ($\rho_u = \rho_l$).

Table K.1: Results for sampling policies for $\mathcal{P}_{\text{Mix}}$ (CIFAR-10 LT, semi-supervised) $\rho = 100$.

| Method | Mean Recall | Min Coverage | Min Recall | Mean Recall |
|---|---|---|---|---|
| Uniform Policy | 83.3 | 0.072 | 70.5 | 83.3 |
| Greedy Policy | 83.6 | 0.093 | 78.2 | 81.8 |
| SelMix Policy | **84.9** | **0.094** | **79.1** | **84.1** |

Table K.2: Comparison of finetuning the feature extractor vs only the linear classification layer (CIFAR-10 LT, semi-supervised) $\rho = 100$

| Method | Mean Recall | Min Coverage | Min Recall | Mean Recall |
|---|---|---|---|---|
| Frozen $g$ | 83.5 | 0.089 | 77.3 | 84.1 |
| Finetuning $g$ | **85.4** | **0.095** | **79.1** | **84.5** |

we find in Table 4 SelMix is able to improve over SotA methods across objectives. Further, SelMix has the same time complexity as CSST (Rangwani et al., 2022) baseline. (Refer Sec. L & Sec. B).

**b) Comparison of Sampling Policies.** We compare the sampling policies ($\mathcal{P}_{\text{Mix}}$): the uniform mixup policy, the greedy policy of selecting the max gain mixup and the SelMix policy (Table K.1). We find that other policies in comparison to SelMix are unstable and lead to inferior results. We compare furtherthe performance of a range of sampling distribution by varying the inverse distribution temperature $s$ in $\mathcal{P}_{\text{SelMix}}$. We find that intermediate values of $s = 10$ work better in practice (Fig. K.1).

**c) Fine-Tuning.** In Table K.2 we observe that fine-tuning $g$ leads to improved results in comparison to keeping it frozen. We further show in App. L that fine-tuning with SelMix is computationally cheap compared to CSST for optimizing a particular non-decomposable objective.

### K.1 DETAILED PERFORMANCE ANALYSIS OF SELMIX MODELS

As in SelMix, we have provided results for fine-tuned models for optimizing specific metrics on CIFAR-10 ($\rho_l = \rho_u$) in Table 2. In this section, we analyze all these specific models on all other sets of metrics. We tabulate our results in Table K.3. It can be observed that when the model is trained for the particular metric for the diagonal entries, it performs the best on it. Also, we generally find that all models trained through SelMix reasonably perform on other metrics. This demonstrates that the models produced are balanced and fair in general. As a rule of thumb, we would like the users to utilize models trained for constrained objectives as they perform better than others cumulatively.

Table K.3: Values of all metric values for individually optimized runs for CIFAR-10 LT ($\rho_l = \rho_u$)

| Optimized On \ Observed Metric | Mean Rec. | Min. Rec. | HM | GM | Mean Rec./Min Cov. | HM/Min Cov. |
|---|---|---|---|---|---|---|
| Mean Rec. | 85.4 | 77.6 | 85.0 | 85.1 | 85.4/0.089 | 85.0/0.089 |
| Min. Rec. | 84.2 | 79.1 | 84.1 | 84.2 | 84.2/0.091 | 84.1/0.091 |
| HM | 85.3 | 77.7 | 85.1 | 85.2 | 85.3/0.091 | 85.1/0.091 |
| GM | 85.3 | 77.5 | 85.1 | 85.3 | 85.3/0.091 | 85.1/0.091 |
| Mean Rec./Min. Cov. | 85.7 | 75.9 | 84.7 | 84.8 | 85.7/0.095 | 84.7/0.095 |
| HM/Min Cov. | 85.1 | 76.2 | 84.8 | 84.9 | 85.1/0.095 | 84.8/0.095 |

### K.2 COMPARISON BETWEEN FIXMATCH AND FIXMATCH (LA)

We find that using logit-adjusted loss helps in training feature extractors, which perform much superior in comparison to the vanilla FixMatch Algorithm (Table K.4). However, our method SelMix

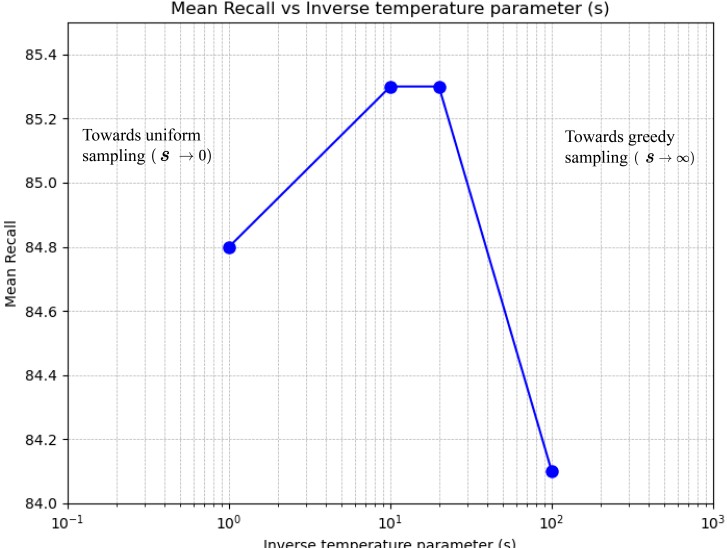

Figure K.1: We show ablation on the inverse temperature parameter ($s$) v/s the performance on the mean recall. For a mean recall optimized run a very small $s$ yields a sampling function close to a uniform sampling, whereas a very large $s$ ends up close to a greedy sampling strategy.

is able to improve both the FixMatch and the FixMatch (LA) variant. We advise users to use the FixMatch (LA) algorithm for better results.

Table K.4: Comparison of the FixMatch and FixMatch (LA) methods with SelMix.

| Method | Mean Recall | Min Coverage | Min Recall | Mean Recall |
|---|---|---|---|---|
| FixMatch | 76.8 | 0.037 | 36.7 | 76.8 |
| w/ SelMix | 84.7 | 0.094 | 78.8 | 82.7 |
| FixMatch (LA) | 82.6 | 0.065 | 63.6 | 82.6 |
| w/ SelMix | 85.4 | 0.095 | 79.1 | 84.1 |

### K.3 VARIANTS OF MIXUP

As SelMix is a distribution on which class samples $(i, j)$ to be mixed up, it can be easily be combined with different variants of mixup (Yun et al., 2019; Kim et al., 2020b). To demonstrate this, we replace the feature mixup that we perform in SelMix, with CutMix and PuzzleMix. Table K.5 contains results for various combinations for optimizing the Mean Recall and Min Recall across cases. We observe that SelMix can optimize the desired metric, even with CutMix and PuzzleMix. However, the feature mixup we performed originally in SelMix works best in comparison to other variants. This establishes the complementarity of SelMix with the different variants of Mixup like CutMix, PuzzleMix, etc., which re-design the procedure of mixing up images.

## L COMPUTATIONAL REQUIREMENTS

The experiments were done on Nvidia A5000 GPU (24 GB). While the fine-tuning was done on a single A5000, the pre-training was done using PyTorch data parallel on 4XA5000. The pre-training was done until no significant change in metrics was observed and the fine-tuning was done for 10k steps of SGD with a validation step every 50 steps. A major advantage of SelMix over CSST is that the process of training a model optimized for a specific objective requires end to end training

Table K.5: Comparison of SelMix when applied to various Mixup variants.

| Method | Mean
*Recall* | Min
*Recall* |
|---|---|---|
| FixMatch | $79.7_{\pm 0.6}$ | $55.9_{\pm 1.9}$ |
| w/ SelMix (CutMix) | $84.8_{\pm 0.2}$ | $75.3_{\pm 0.1}$ |
| w/ SelMix (PuzzleMix) | $85.1_{\pm 0.3}$ | $75.2_{\pm 0.1}$ |
| w/ SelMix (Features-Ours) | $85.4_{\pm 0.1}$ | $79.1_{\pm 0.1}$ |

Table L.1: Comparison of time taken across datasets, for the calculation of Gain using SelMix (Alg. 1) was done on GPU (NVIDIA RTX A5000).

| Dataset | CIFAR-10 LT ($\rho = 100$) | (CIFAR-100 LT $\rho = 100$) | Imagenet-1k LT |
|---|---|---|---|
| | 0.02 sec. | 1.3 sec. | 124 sec. |

which is computationally expensive($\sim$10 hrs on CIFAR datasets). Our finetuning method takes a fraction ($\sim$1hr on CIFAR datasets) of what it requires in computing time compared to CSST. An analysis for computing the Gain through Alg. 1, is provided in Table L.1. We observe that even for the ImageNet-1k dataset, the gain calculation doesn't require large amount of GPU time. Further, an efficient parallel implementation across classes can further reduce time significantly.

## M   LIMITATION OF OUR WORK

In our current work, we mostly focus on our algorithm's correctness and empirical validity of SelMix across datasets. Another direction that could be further pursued is improving the algorithm's performance by efficiently parallelizing the operations across GPU cores, as the operations for each class are independent of each other. The other direction for future work could be characterization of the classifier obtained through SelMix, using a generalization bound. Existing work (Zhang et al., 2021) on the mixup method for accuracy optimization showed that learning with the vicinal risk minimization using mixup leads to a better generalization error bound than the empirical risk minimization. It would be interesting future work to show a similar result for SelMix.

## N   ADDITIONAL RELATED WORKS

**Selective Mixup for Robustness.** The paper SelecMix (Hwang et al., 2022) creates samples for robust learning in the presence of bias by mixing up samples with conflicting features, with samples with bias-aligned features. The paper demonstrates that learning on these samples leads to improved out-of-distribution accuracy, even in the presence of label noise. This paper selects the samples to mixup based on the similarity of the labels, to improve mixup performance on regression tasks. Existing works (Hwang et al., 2022; Yao et al., 2022; Palakkadavath et al., 2022) show that mixups help train classifiers with better domain generalization. Teney et al. (2023) show that resampling-based techniques often come close in performance to mixup-based methods when utilizing the implicit sampling technique used in these methods. It has been shown that mixup improves the feature extractor and can also be used to train more robust classifiers (Hwang et al., 2022). Palakkadavath et al. (2022) show that it is possible to generalize better to unknown domains by making the model's feature extractor invariant to both variation in the domain and any interpolation of a sample with similar semantics but different domains.

**Black-Box Optimization.** We further note that there are some recent works (Li et al., 2023; Wierstra et al., 2014) which aim to fine-tune a model based on local target data for an specific objective, however these methods operate in a black-box setup whereas SelMix works in a white-box setup with full access to model and its gradients.

