# OpenReview forum: "Selective Mixup Fine-Tuning for Optimizing Non-Decomposable Objectives"
_ICLR.cc/2024/Conference — ICLR 2024 spotlight_

### Official Review · Reviewer_4axA · 2023-10-19

**Soundness:** 3 good
**Presentation:** 3 good
**Contribution:** 3 good
**Rating:** 6
**Confidence:** 3

**Summary:**

The authors propose SelMix which utilizes a pre-trained model for representations and optimizes it for improving the desired non-decomposable objective through fine-tuning. They developed a selective sampling distribution on class samples to selectively mix up, optimizing the given non-decomposable objective. The distribution is also updated periodically based on feedback from the validation set. The efficacy of SelMis is evaluated across a wide range of linear or non-linear objectives under both supervised and semi-supervised settings.

**Strengths:**

1. The key novelty lies in the idea of selective mixup. The selective procedure ensures that at each timestep the the objective is optimized.

2. The theoretical analyses present in the paper are sufficient. They provide the convergence analysis and show the validity of the sampling procedure.

3. The experimental results are comprehensive, which consist of strong empirical results under different supervision settings and different distributions.

**Weaknesses:**

1. The authors use the first-order Taylor expansion on Eq. 5. I am curious about the performance if the second-order terms are involved, i.e., calculating the third term of Eq. 5. Also, the first-order gradient can be also approximated by f(x+ h) + f(x-h) -f(x) / 2h. Would it give better performance than the current one-sided approximation?

2. Can authors provide some ablation studies on the hyper-parameters, such as the imbalanced ratio and the number of samples for feedback?

**Questions:**

Please see weaknesses.

---

> ### Author Response · Authors · 2023-11-20
> **Response to Reviewer 4axA [1/2]**
>
> We sincerely thank reviewer 4axA for their thoughtful questions and efforts to review our work. We provide our clarifications below to the concerns mentioned.
>
> > The authors use the first-order Taylor expansion on Eq. 5. I am curious about the performance if the second-order terms are involved, i.e., calculating the third term of Eq. 5. Also, the first-order gradient can be also approximated by f(x+ h) + f(x-h) -f(x) / 2h. Would it give better performance than the current one-sided approximation?
>
> As long as we consider a first-order method similar to SGD, it is sufficient to consider the first-order term in eq (5). Further, considering any second-order terms will involve the calculation of Hessian to take advantage in optimization, calculation of that *which is expensive for neural networks* with large $W$.
>
> We now explain the above in more detail. By considering the second-order term in eq (5), the definition of gains would slightly change, and the expected directional vector $V = \sum_{i,j}P_{Mix} (i, j) V_{ij}$ would slightly change accordingly.
> On the other hand, in the convergence analysis (Theorem 4.2), the expected directional vector affects the constant $c$, which is a lower bound of the cosine similarity of the expected directional vector and the true (normalized) gradient.
> If the cosine similarity is larger, then we have better convergence.
> Since maximizing the cosine similarity is equivalent to maximizing Gain, our method already aims to maximize the cosine similarity.
> This indicates that when we consider the second-ordered term in eq (5), to achieve better performance, we have to consider a second-order method using a Hessian of the loss function eq (2) instead of SGD. Therefore, as long as we consider a first-order method such as SGD, it is sufficient to consider the first-order term in eq (5).
>
> Regarding the finite difference $(f(x + h) - f(x - h))/2h$, to the best of our knowledge, such an estimation is used for an estimation of gradients of the smoothed version of the objective function (e.g., [Agarwal el al., 2010]).
> Since we assume that the change (i.e., Gain) of the objective $\psi$ can be approximated as
> a linear function of the directional vector $V$, we do not see an advantage of considering the two sided-approximation.
> More precisely, we assume that the change
> $\Delta \psi(V) := \psi(\widetilde{C}[(W + \eta V)^T g]) - \psi(\widetilde{C}[W^T g])$
> is approximately equal to a linear function of $V$,
> where $V$ is a directional vector and $\eta > 0$ is a learning rate.
> Since $\Delta \psi(V)$ is approximately linear in $V$,
>  the two side approximation $\Delta \psi(V) - \Delta\psi(-V)$ is approximately equal to $2 \Delta \psi(V)$.
> Therefore, by considering two-sided approximation, we would only have an additional constant factor (2) in our argument.
> However, in a more generalized setting, where
> the assumption regarding the linear approximation does not hold,
> we might be able to improve our method (e.g. in terms of robustness) by utilizing a two-sided approximation. This would be an interesting extension.
>
> > Can authors provide some ablation studies on the hyper-parameters, such as the imbalanced ratio and the number of samples for feedback?
>
>  We request you look at Appendix K, where we have provided analysis on hyper-parameters. Here, we provide the ablation of the size of the validation set vs. the Mean Recall for the CIFAR-10 dataset with an imbalance ratio, $\rho_l=\rho_u=100$, where we observe robust performance of SelMix even when data is reduced to 10% of initial.
>
> **Table 2: Comparison of SelMix's performance for different sizes of the feedback set.**
> | Percentage of <br> samples used | Mean Recall |
> |-|-|
> | 5k                     | 85.5        |
> | 2.5k                     | 84.8        |
> | 1.25k                       | 84.7        |
> | 0.5k                       | 84.4        |
>
> We show the ability of SelMix to boost performance in various imbalanced settings. We consider the CIFAR-10 dataset with an imbalance ratio, $\rho_l=\rho_u=50, 100, 200$ and show the results for optimizing Mean Recall and Min. Recall.

---

> ### Author Response · Authors · 2023-11-20
> **Response to Reviewer 4axA [2/2]**
>
> **Table 2: Comparison of SelMix's performance on different imbalance factors (Setup as in Table 1).**
>
> | Method       |   Mean Recall  $(\rho_l = \rho_u=50)$ |  Min Recall $(\rho_l = \rho_u=50)$ |  Mean Recall $(\rho_l = \rho_u=100)$ |   Min Recall $(\rho_l = \rho_u=100)$ |   Mean Recall $(\rho_l = \rho_u=200)$ |   Min Recall $(\rho_l = \rho_u = 200)$ |
> |-|-|-|-|-|-|-|
> | FixMatch(LA) | 84.5                            | 73.8                           | 79.7                             | 55.9                          | 74.3                             | 33.8                          |
> | w/ SelMix    | 87.1                            | 81.1                           | 85.4                             | 79.1                          | 81.6                             | 64.6                          |
>
> Further in the Appendix, we compare the performance of different sampling policies for different values of scaling parameters (Fig. K.1). We also compare against different substitute mixups for the input, such as PuzzleMix and CutMix (Tab. K.5). We have provided results for different permutations of imbalance ratios for the labeled and unlabeled data such as $\rho_l = \rho_u = 100$ in Tab. 2(L) , $(\rho_l, \rho_u)= (100, 1)$ and $(\rho_l, \rho_u) = (100, \frac{1}{100})$ in Appendix Tab. H.1
> We also compare the effectiveness of substituting cross-entropy loss with LA loss in FixMatch, which yields a better feature extractor, resulting in an overall boost in performance in Tab. K.4.
>
> We hope our response has clarified your concerns; please feel free to reach out to us if there are any other questions.
>
>
>
> ## References
> 1. Agarwal, Alekh, Ofer Dekel, and Lin Xiao. "Optimal Algorithms for Online Convex Optimization with Multi-Point Bandit Feedback." Colt. 2010.

---

### Official Review · Reviewer_LrpR · 2023-10-25

**Soundness:** 3 good
**Presentation:** 4 excellent
**Contribution:** 3 good
**Rating:** 8
**Confidence:** 3

**Summary:**

The paper introduces a novel approach called Selective Mixup (SelMix), which leverages an optimized sampling distribution of elements to mix based on a specific objective. This sampling distribution is designed to consider the gain in the objective function achieved by mixing centroids of class samples. Throughout the training process, the sampling distribution is dynamically updated to adapt to changes in class centroids. The paper also includes a rigorous theoretical analysis, establishing the approach's convergence rate and the validity of the sampling distribution. To validate its effectiveness, the method is empirically evaluated across various long-tailed benchmark datasets in both supervised and semi-supervised scenarios, by fine-tuning pretrained models using SelMix.

**Strengths:**

- **Theoretical Analysis and Convergence Rate:** The paper's presents a rigorous theoretical analysis, particularly the establishment of a convergence rate. This analysis adds credibility and reliability to the proposed SelMix.
- **Strong Experimental Results:** The paper showcases strong experimental results compared to other state-of-the-art approaches. The empirical evaluation is performed across different long-tailed benchmark datasets in both supervised and semi-supervised settings, and for different objectives.
- **Efficiency of the proposed approach:** As discussed in Appendix L, the paper is much more computationally efficient than previous methods, since it relies on fine-tuning a pretrained model.

**Weaknesses:**

- **On a Fair Comparison with Other Methods:** The paper uses FixMatch + Logits Adjusted (LA) loss as a baseline, while other approaches are evaluated using the vanilla FixMatch. This discrepancy in the comparison may not provide a fair assessment of the true impact of SelMix. As can be seen in Table H.2 and H.1, FixMatch + LA shows already significant improvements over vanilla FixMatch. This makes it challenging to evaluate whether SelMix's improvements reported in the benchmarks are due to the approach or its combination with a more advanced baseline. A fair comparison would involve evaluating SelMix with vanilla FixMatch to better understand its relative performance, or comparing to other state-of-the-art methods when using FixMatch + LA.
- **Lack of Discussion about other related "Selective Mixup":** Several recent paper have discussed using *selected pairs* of examples to mix, between specific classes or different domains [1,2,3,4,5]. Even though the approach presented is applied in a different context and quite novel compared to these papers, I think a discussion about the relation to these methods is important since the paper is currently part of a popular topic.

[1]: Yao H. et al., C-mixup: Improving generalization in regression. In NeurIPS 2022
[2]: Yao H. et al., Improving out-of-distribution robustness via selective augmentation. In ICML 2022
[3]: Hwang I. et al., Selecmix: Debiased learning by contradicting-pair sampling. In NeurIPS 2022
[4]: Palakkadavath R. et al., Improving domain generalization with interpolation robustness. In NeurIPS 2022 Workshop on Distribution Shifts: Connecting Methods and Applications
[5]: Teney D. et al., Selective mixup helps with distribution shifts, but not (only) because of mixup. In arXiv:2305.16817, 2023.

**Questions:**

- What is the *prior distribution* on labels $\pi_i$ ? How do you define this in practice ?
- What is the *unconstrained confusion matrix* $\tilde{C}$ ? It is never clearly defined in the paper, but is important to define the gain.
- I would like to see at least some comparison of SelMix using "vanilla" FixMatch to fairly compare with state of the art.

---

> ### Author Response · Authors · 2023-11-20
> **Response to Reviewer LrpR [1/2]**
>
> We thank the reviewer LrpR for their efforts in reviewing our work and providing us with very valuable feedback. We address the concerns raised by you as follows.
>
> > This discrepancy in the comparison may not provide a fair assessment of the true impact of SelMix. As can be seen in Table H.2 and H.1, FixMatch + LA shows already significant improvements over vanilla FixMatch. This makes it challenging to evaluate whether SelMix's improvements reported in the benchmarks are due to the approach or its combination with a more advanced baseline. A fair comparison would involve evaluating SelMix with vanilla FixMatch to better understand its relative performance,
>
>
>
> Thanks for pointing out this inconsistency due to **a typo of w/**; we have corrected typos in the revision. We want to clarify that all the methods we have compared in Table H.1 are methods **which are trained from scratch** and are not finetuning methods like the proposed SelMix (hence, w/ is not correct). Further, while comparing, we made sure that we were using the **best baselines** for proposed methods; for example, we used LA for DASO as their implementation as their implementation (https://github.com/ytaek-oh/daso) recommended it. Hence, to the best of our knowledge, we have **provided fair comparisons**.
>
> > Results with FixMatch
>
> One potential work that uses a similar framework is CoSSL, which is a two-stage framework comprising a pre-training and a fine-tuning stage. For performing fair comparison with SelMix finetuning, we use the same pre-trained model for both for the results below. It can be observed that SelMix finetuning outperforms CoSSL fine-tuning by a large margin.
>
>
>
>
> **Dataset: CIFAR-10, IBR_L = 100, IBR_U = 100, N1=1500, M1=3000** (Table 1)
> |                     | Mean Rec.         | Min Rec.          | HM                | GM                | Mean Rec./Min Cov |                   | Mean Rec.         | Min Rec.          | HM                | GM                | Mean Rec./Min Cov |
> |-|-|-|-|-|-|-|-|-|-|-|-|
> | FixMatch(LA)        | 79.7              | 55.9              | 76.7              | 78.3              | 79.7/0.056        | FixMatch           | 76.9              | 43.5              | 71.9              | 74.5              | 76.9/0.044        |
> | FixMatch(LA) + CoSSL | 81.8              | 72.3              | 81.1              | 81.5              | 81.8/0.078        | FixMatch + CoSSL  | 82.6              | 72.0              | 81.9              | 82.1              | 82.6/0.076        |
> | FixMatch(LA) + SelMix | 85.4              | 79.1              | 85.1              | 85.3              | 85.7/0.095        | FixMatch + SelMix | 85.4              | 79.5              | 84.2              | 84.3              | 85.4/0.096        |
>
>
> **Dataset: CIFAR-10, IBR_L = 100, IBR_U = 1, N1=1500, M1=3000** (Table H.1)
> |                     | Mean Rec.         | Min Rec.          | HM                | GM                | Mean Rec./Min Cov |                   | Mean Rec.         | Min Rec.          | HM                | GM                | Mean Rec./Min Cov |
> |-|-|-|-|-|-|-|-|-|-|-|-|
> | FixMatch(LA)        | 90.1              | 75.8              | 89.5              | 89.7              | 90.1/0.083        | FixMatch           | 82.8              | 59.1              | 80.6              | 82.3              | 82.8/0.059        |
> | FixMatch(LA) + CoSSL | 89.9              | 75.9              | 89.4              | 89.7              | 89.9/0.085        | FixMatch + CoSSL  | 87.5              | 73.3              | 86.8              | 87.2              | 87.5/0.075        |
> | FixMatch(LA) + SelMix | 91.4              | 84.7              | 91.3              | 91.3              | 91.4/0.096        | FixMatch + SelMix | 87.9              | 82.3              | 87.4              | 87.6              | 87.9/0.094        |
>
>
>
>
> **Dataset: CIFAR-10, IBR_L = 100, IBR_U = 0.01, N1=1500, M1=3000** (Table H.1)
>
> |                     | Mean Rec.         | Min Rec.          | HM                | GM                | Mean Rec./Min Cov |                   | Mean Rec.         | Min Rec.          | HM                | GM                | Mean Rec./Min Cov |
> |-|-|-|-|-|-|-|-|-|-|-|-|
> | FixMatch(LA)        | 75.5              | 45.1              | 71.1              | 73.3              | 75.5/0.046        | FixMatch           | 71.3              | 28.5              | 61.3              | 67.1              | 71.3/0.030        |
> | FixMatch(LA) + CoSSL | 77.3              | 64.3              | 75.8              | 76.5              | 77.3/0.070        | FixMatch + CoSSL  | 77.5              | 63.3              | 75.9              | 76.6              | 77.5/0.070        |
> | FixMatch(LA) + SelMix | 81.3              | 74.3              | 81.0              | 80.9              | 81.7/0.091        | FixMatch + SelMix | 79.8              | 72.6              | 78.9              | 79.1   | 79.8/0.082        |

---

> ### Author Response · Authors · 2023-11-20
> **Response to Reviewer LrpR [2/2]**
>
> > What is the prior distribution on labels? How do you define this in practice ?
>
> Prior distribution $\pi_i$ over labels $y=1...K$, represents the probabilities of obtaining a sample with a label $y=i$when sampled at random from a dataset before observing any specific data.
> We have obtained the prior distribution estimates, using the label statistics $y$ of the labeled training data present.
>
> > What is the unconstrained confusion matrix? It is never clearly defined in the paper, but is important to define the gain.
>
> Thanks for the interesting question, it actually captures mathematical beauty of the proposed approximation.
> The unconstrained confusion matrix $\widetilde{C}$ is implicitly defined in the "Estimation of Gain Matrix" paragraph by $C_ {i} = \pi_ {i} softmax(\widetilde{C}_ {i})$, where $C_ {i}$ and $\widetilde{C}_ {i}$ are $i$-th row vectors of
> $C$ and $\widetilde{C}$, respectively.
> Although the matrix $\widetilde{C}$ is not uniquely determined since the softmax function is not one-to-one,
> the approximation (the first term of the RHS of eq (8)) is unique as the derivative of all these are unique.
> We explain this more in detail.
> In the approximation formula, we only need jacobian
> $\partial \psi/\partial \widetilde{C} = \frac{\partial \psi}{\partial {C}} \frac{\partial {C}}{\partial{\widetilde{C}}}$.
> We note that gradient of the softmax function can also be written as a function of the softmax function (i.e., $\frac{\partial \sigma_l}{\partial \xi_m} = \delta_{lm} \sigma_l(\xi) - \sigma_l(\xi) \sigma_m(\xi)$, where $\sigma_l(\xi) = softmax_l(\xi)$
> for $\xi \in \mathbb{R}^K$, $1 \le l, m \le K$).
> Therefore, the first term of the RHS of eq (8) is uniquely determined even if $\widetilde{C}$ is not uniquely determined.
> We appreciate your suggestion and will clarify this in the final version.
>
> We sincerely hope that our response has cleared the misunderstanding due to a type, which leads to an improved impression of our paper. Please feel free to contact us for further clarifications.

---

> > ### Comment · Reviewer_LrpR · 2023-11-21
> > **Thanks for the answer**
> >
> > Thanks to the authors for their detailed answer.
> > - The three tables presented clearly show that the improvements are not limited to the use of Fixmatch + LA. This clears my concerns about the fair comparison. Thanks for taking the time to do the additional experiments.
> > - I better understand the role of $\tilde{C}$. The proposed approximation is indeed interesting and deserves to be emphasized more. It was not clear from my first read of the paper, so I encourage the authors to clarify this in the final version.
> >
> > As all my concerns have been addressed, I'm raising my score to 8 (accept).

---

### Official Review · Reviewer_iMBB · 2023-11-01

**Soundness:** 3 good
**Presentation:** 3 good
**Contribution:** 4 excellent
**Rating:** 8
**Confidence:** 4

**Summary:**

For applications with critical consequences, accuracy is not a suitable performance metric, and other metrics such as recall h-mean and worst-case recall should be used. However, such metrics are non-decomposable, which means that they cannot be expressed as a simple average of a function of label and prediction pairs calculated for each sample. Prior techniques to optimize non-decomposable objectives such as CSST lead to sub-optimal representations. Other methods to improve the performance on long-tailed class-imbalanced datasets such as DASO, ABC, and CoSSL perform suboptimally for the non-decomposable objectives. This paper proposes SelMix -- a technique that utilizes a pre-trained model for representations and optimizes it for improving the desired non-decomposable objective through fine-tuning.

**Strengths:**

In the experimental results for matched label distributions shown in Table 2, the proposed method shows shows superior performance compared to existing methods; DARP, CReST, CReST+, ABC, CoSSL, DASO, and CSST for all metrics; mean recall, min recall, H-mean, G-mean, and min coverage. The results for unknown label distributions in Figure 3 also shows that the proposed method outperforms existing methods on all metrics. The same is true for large datasets such as ImageNet-1k LT, shown in Table 4.

**Weaknesses:**

The fact that the proposed method requires only fine-tuning of a pre-trained model should result in a huge advantage in training time compared to existing methods, but this is not highlighted in the main results section of the paper. The only mention of computational requirements is in Appendix L, where only the time to calculate the Gain is shown. It would be interesting to see a more comprehensive comparison of the training time of the proposed method vs. all the existing methods.

**Questions:**

The description of SelMix in Algorithm 1 is a bit confusing. Algorithm 1 says the classifier h is updated, but in Section 4.1 it says the parameter W is updated. I’m assuming the former is a consequence of the latter. If this is the case, wouldn’t it be better to replace h with W in Algorithm 1? Obfuscating the actual operation with the function SGD-Update() also makes it difficult to see the exact algorithm.

---

> ### Author Response · Authors · 2023-11-20
> **Response to Reviewer iMBB**
>
> We thank the reviewer iMBB for their reviews. We are delighted to know that you appreciate various aspects of our work. Thank you! We address some of the concerns raised by you as follows.
>
> > The fact that the proposed method requires only fine-tuning of a pre-trained model should result in a huge advantage in training time compared to existing methods, but this is not highlighted in the main results section of the paper.
>
> We agree. Our method is a finetuning method, and in order to optimize for a performance objective, we do not have to train from scratch.  We compare our method training time with CSST which needs to train the model from scratch for each objective. However, for this analysis, as CSST does not support the full list of objectives in the table; we use the closest surrogate object was used. The pretraining and training for SelMix and CSST was done on 4XA5000 (24gb) gpus.
>
> **Table 1.** Time comparison for SelMix with CSSTDataset: CIFAR-10, $\rho_l$ = $\rho_u$ = 100, $N_1$=1500, $M_1$=3000
>
> | Method              | CSST | SelMix |
> | ------------------- | ---- | ------ |
> | Pretraining         |  NA      |    20hrs 40m   |
> | Mean Rec.           |  21 h    |    35 m    |
> | Min Rec.            |  21 h    |    35 m    |
> | HM                  |  21 h    |    35 m    |
> | GM                  |  21 h    |    35 m    |
> | Mean Rec. / Min Cov |  21 h    |    35 m    |
> | Total time          |  105 h   |    23h 35m   |
>
> We will emphasize this more in the final version of the draft; thanks for your suggestion.
>
> > The description of SelMix in Algorithm 1 is a bit confusing. Algorithm 1 says the classifier h is updated, but in Section 4.1 it says the parameter W is updated. I’m assuming the former is a consequence of the latter. If this is the case, wouldn’t it be better to replace h with W in Algorithm 1?
>
>
> Thank you for your suggestion. Yes, you are correct that parameter $W$ is the one which is primarily updated by the algorithm. However, in practice, a small amount of fine-tuning of the feature extractor also helps our case. Hence, we have named our algorithm SelMix as a fine-tuning technique and used $h$ in the algorithm (which contains $W$ as the primary part). We have discussed this in Appendix K of the draft.
>
>
> > Obfuscating the actual operation with the function SGD-Update() also makes it difficult to see the exact algorithm
>
> We appreciate your suggestion. Due to space constraints, we had to abstract out the function. We have provided the full-length SelMix Algorithm in App. Alg. 2, which we will add to final version with increased page limit.

---

### Official Review · Reviewer_misU · 2023-11-01

**Soundness:** 3 good
**Presentation:** 2 fair
**Contribution:** 3 good
**Rating:** 6
**Confidence:** 4

**Summary:**

This paper aims to fine-tune a pre-trained model using an additional objective, especially a non-decomposable objective. In particular, the authors propose a mixup-based technique that can determine a sampling distribution over classes for performing objective-oriented mix-up. Empirical results demonstrate the efficacy of the proposed method.

**Strengths:**

1. The proposed method, SelMix, is simple and reasonable, with a theoretical guarantee.
2. Sufficient empirical results are provided.

**Weaknesses:**

1. Literature review is not sufficient, and some sentences are over-claimed. Fine-tuning a pretrained network for non-decomposable objectives and even non-differential objectives has been extensively developed recently. Please refer to a recent paper [1] and the references therein for more details. Particularly, the objective derived $\mathbb{E}[G]=\sum_{i, j} G_{i, j} \mathcal{P}_{M i x}(i, \jmath)$ shares the same motivation as the natural evolution strategies [2] adopted in [1].
2. There are many grammar errors, needed to double check. For example, "existing frameworks theoretical frameworks", "Semi-Supervised Learning is are algorithms", $f: \mathbb{R}^{m \times n} \rightarrow n$, $\rho_l=N_1 / N_K, \rho_l=M_1 / M_K$.
3. Some theorems are not formal and have overly strong assumptions. For instance, $z_k$ in Theorem 4.1 appears suddenly without intuitive explanation. There is no detailed explanation in the proof in the Appendix. “a reasonable directional vector for optimization” is not professional.

[1] Li, Jing, et al. "Earning Extra Performance from Restrictive Feedbacks." IEEE Transactions on Pattern Analysis and Machine Intelligence (2023).
[2] Wierstra, Daan, et al. "Natural evolution strategies." The Journal of Machine Learning Research 15.1 (2014): 949-980.

**Questions:**

1.	What is the difference between "Non-Decomposable Objective" and "Non-Differentiable Objective"?
2.	The first-order Taylor approximation is adopted in this paper for calculating the Gain matrix, which cannot guarantee convergence for complex objectives, although it is simple and efficient.
3.	It seems we can fitting a surrogate model to approximate the Non-Decomposable Objective, which can then be used for gradient-based model fine-tuning. It would be interesting to discuss the advantages of the first-order Taylor approximation compared to fitting a surrogate model.

---

> ### Author Response · Authors · 2023-11-20
> **Response to Reviewer misU [1/3]**
>
> We sincerely thank reviewer misU for their constructive feedback on our work. We address your concerns as follows.
> > Literature review is not sufficient, and some sentences are over-claimed. Fine-tuning a pre-trained network for non-decomposable objectives and even non-differential objectives has been extensively developed recently. Please refer to a recent paper [1] and the references therein for more details. Particularly, the objective derived
>  shares the same motivation as the natural evolution strategies [2] adopted in [1].
>
> We thank the reviewer for pointing towards a concurrent paper [1], which also tries to fine-tune a model, but it operates in the scenario where the feedback is only accesssible through queries (**black-box**), whereas *our work* SelMix works in a scenario where model gradients are not-computable for certain objectives but can be approximated with gradients on other objectives (**white-box**). We have cited the papers in the additional related work section of our revision. Since the method proposed in [1] is based on a black-box optimization method proposed in [2],
> in [1] values of the objective $\psi$ can be observed only through function query, whereas
> we assume the objective $\psi$ and its gradients w.r.t the confusion matrix $C$ are approximately known.
> **Since our method is a white-box optimization method, it can converge faster than black-box optimization methods such as [1]**.
> Because we were not able to find convergence analysis in [1], we compare our result to a standard result [Agarwal et al, 2010]
> in the black-box optimization (bandit) literature.
> Corollary 4 in [Agarwal et al, 2010] suggests that the convergence rate of the black box optimization method is $O(d^2 K^2/\sqrt{t})$,
> while our convergence rate in Theorem 4.2 is given as $O(1/t)$,
> which indicates that our approach is less prone to larger values of $d$ and $K$,
> thus, our approach achieves better values of the objective compared to the black-box optimization approach [1].
>
>
> > "Some theorems are not formal and have overly strong assumptions. For instance,
> $z_k$ in Theorem 4.1 appears suddenly without intuitive explanation. There is no detailed explanation in the proof in the Appendix. “a reasonable directional vector for optimization” is not professional."
>
> Thank you for your suggestion. We have improved our paper based on your comments (please refer to the submitted revision).
>
>
> Although statements of Theorem 4.1 and 4.2 are formal, due to space constraint the statement of Theorem 4.3 is informal. The formal statement of Theorem 4.3 can be found in Section C.4.
>
> The mean feature vector $z_i$ is defined in Theorem 4.1.
> We provide an intuitive explanation why this is necessary.
> Please note that $\psi$ is a function of the confusion matrix $C[h]$ of the classifier $h(x) = W^T g(x)$ (Sec. 3).
> By definition the $(i, j)$-entry of the confusion matrix $C[h]$ is given as
>
> $$
>       \mathbf{E}_ {x, y \sim D}[ 1 (y=i, \mathrm{argmax}_ {l} h(x)_ {l} = j) ]=
>        P(y=i) \mathbf{E}_ {x \sim P(\cdot | y=i)}[ 1(\mathrm{argmax}_ {l} h(x)_ {l} = j) ]
> $$
>
> Here, $P(\cdot | y = i)$ is the class conditional probability distribution.
> If we consider softmax instead of argmax, $C_ {ij}[h]$ is approximately given as
>
> $$
>      C_ {ij}[h] \approx P(y=i) \mathbf{E}_ {x \sim P(\cdot | y=i)}[ \mathrm{softmax}_ {l} h(x)].
> $$
>
> Since we are interested in the change in the objective $\psi$ (Gain), by the above formula, we are interested in the change in the mean of the softmax
> of $h(x)$, where $x$ belongs to the class $i$.
> Because $h(x) = W^T g(x)$ is a linear classifier, the change in the mean of the softmax is related to the mean feature vector
> $z_i = \mathbf{E}_ x [g(x)]$, where $x$ belongs to the class $i$.
>
> In Sec. 4.1, the words “a reasonable directional vector for optimization” appear twice.
> The first appearance follows the following inequality and “a reasonable directional vector for optimization.”
> is an intuitive explanation of this inequality for the vector $\widetilde{V}^{(t)}$. We have modified it to "a vector which has a sufficient component in the direction of gradient" for optimization.
>
> $$
> \mathbf{E}[\widetilde{V}^{(t)}] \cdot
>         \frac{\partial \psi(W^{(t)}) }{\partial W}
>          > c\|\frac{\partial \psi(W^{(t)}) }{\partial W}\|.
> $$

---

> > ### Author Response · Authors · 2023-11-20
> > **Response to Reviewer misU [2/3]**
> >
> > > What is the difference between "Non-Decomposable Objective" and "Non-Differentiable Objective"?
> >
> > We would like to clarify that both "Non-Decomposable Objectives" and "Non-Differentiable Objectives" are two orthogonal categories. A differentiable objective is characterized by its ability to have a computable derivative with respect to the model's parameters and vice-versa. For instance, the $L_2$ error serves as a prime example of a differentiable objective commonly employed in regression tasks. In the realm of classification, the majority of objectives are non-differentiable, owing to their reliance on the indicator function $I_{y=y'}$. This function introduces non-differentiability due to its discrete nature. On the other hand, a decomposable objective is one that can be expressed as the mean or sum of smaller objectives. In the context of a classifier, these smaller objectives often hinge on the label-prediction pairs, as expressed by the following formula: $\psi  = \frac{1}{N} \sum_{i=1..N}F(y,y')$. Accuracy is a decomposable objective since $Acc = \frac{1}{N} \sum_{i=1..N}I_{y=y'}$ where $I_{y=y'}$ is the accuracy for the individual sample, yet it is non-differentiable. Although most useful objectives are both non-decomposable and non-differentiable, one does not imply the other as shown in the following table.
> >
> > Table 1: Examples of objectives based on their decomposability and differentiability
> > |  | Decomposable | Non-Decomposable |
> > |----------|----------|----------|
> > | **Differentiable** | Mean $L_2$ error | H-mean of individual $L_2$ errors |
> > | **Non-differentiable** | Accuracy |  G-mean of Recall |
> >
> >
> >
> > > The first-order Taylor approximation is adopted in this paper for calculating the Gain matrix, which cannot guarantee convergence for complex objectives, although it is simple and efficient.
> >
> > We note that using the first-order approximation is sufficient for first order methods such as SGD (we refer to the answer to reviewer 4axA).
> > At the cost of slower convergence rate, One can optimize the objective under more generalized setting without using the first-order Taylor approximation.
> > More specifically, in a black-box optimization setting,
> > Table I in [Scarlett et al., 2017] suggests
> > that we need at least $\Omega(\log^{dK/2} (1/\varepsilon))$ times more iterations
> > to achieve $\varepsilon$-optimal solution
> > compared to the first order method (in the worst case).
> >
> >
> > > It seems we can fitting a surrogate model to approximate the Non-Decomposable Objective, which can then be used for gradient-based model fine-tuning. It would be interesting to discuss the advantages of the first-order Taylor approximation compared to fitting a surrogate model.
> >
> > Thanks for an interesting question. We want to convey that all existing heuristic-based techniques like ABC, DASO, etc. utilize a surrogate loss to approximate the performance of the models on LT data.
> > For instance, we note that the loss function considered by ABC (Lee et al., 2021) can be regarded as a surrogate loss for the mean recall across classes (we explain this more in detail in the "loss proposed by ABC" paragraph below).
> >
> > We have shown in Table 1 and Fig. 3 *that our proposed method, SelMix, outperforms these heuristic surrogate-based methods (ABC, DASO etc.)*.
> >
> > Further, developing a surrogate function for each metric is hard, whereas using any metric with SelMix is straightforward.
> >
> > In the SelMix framework, it is easy to combine constraints, like coverage, into an optimization problem. Whereas it is hard to come up with surrogates which also incorporate constraints.
> >
> > We will discuss these in the final version of the draft; thank you for the suggestions.

---

> ### Author Response · Authors · 2023-11-20
> **Response to Reviewer misU [3/3]**
>
> **Loss Proposed by ABC**
>
> ABC (Lee et al., 2021) considers the loss function of the following form for each example $(x, y)$ (Eq. 2).
> Here, for $i = 1, \dots, K$, $N_i$ is the number of samples in class $i$ (the class $K$ is the tail class).
>
> $$
>       L(F(x), y) = B (\frac{N_ {K}}{N_ {y}}) CE (F(x), y).
> $$
>
> Here, for $p \in (0, 1)$, $B(p)$ is a Bernoulli Random Variable with $E[B(p)] = p$.
> We denote by $D$ the original long-tailed data distribution and by $D_ {B}$ the balanced data distribution
> so that we have the following for any (measurable) function $f$ on the data set:
>
> $$
> E _ {(x, y) \sim D} [f(x, y)] = E_ { (x, y) \sim D_ {B}} [\pi_ {y} f(x, y)].
> $$
>
> Here $\pi_i = P (y=i)$ is the class prior of the long-tailed distribution for class $i$.
>
> In the definition of $L(F(x), y)$, by taking the expectation, we have the following:
>
> $$
> E_ {(x, y) \sim D} [L(F(x), y)] = E_ {(x, y) \sim D_ {B}}[ \frac{N_ {K} \pi_ {y}}{N_ {y}} CE(F(x), y)].
> $$
>
> Since we have approximately $\pi_ {y} \propto N_ {y}$, we see that
>
> $$
> E_ {(x, y) \sim D} [L(F(x), y)]  \approx const \times E_ {(x, y) \sim D_ {B}}[CE(F(x), y)].
> $$
>
> Since under a balanced data distribution, accuracy coincides with the mean recall across classes and the CE loss is a surrogate loss for accuracy, we can regard the ABC loss as a surrogate loss of the mean recall.
>
> We hope our response has clarified your concerns; please feel free to reach out to us if there are any other questions.
>
> ## References
> 1. Agarwal, Alekh, Ofer Dekel, and Lin Xiao. "Optimal Algorithms for Online Convex Optimization with Multi-Point Bandit Feedback." Colt. 2010.
> 2. Scarlett, Jonathan, Ilija Bogunovic, and Volkan Cevher. "Lower bounds on regret for noisy gaussian process bandit optimization." Conference on Learning Theory. PMLR, 2017.

---

### Meta-Review · Area_Chair_kp9H · 2023-12-02

**Metareview:**

The paper addresses the question of optimizing non-decomposable performance objectives by rethinking the sampling distribution via a selective mixup augmentation strategy. Overall, the proposed method is principled and novel, while the conducted experiments validate the research hypotheses of the paper. The reviewers expressed positive opinions in the reviews and the authors provided adequate explanations to the raised questions. I therefore recommend an acceptance of the paper.

**Justification For Why Not Higher Score:**

The paper would need a few further ablations for an oral, and there are some clarity issues in writing, which the reviewers pointed out.

**Justification For Why Not Lower Score:**

The paper has merits and the conducted experiments validate the research hypotheses.

---

### Decision · Program_Chairs · 2024-01-16

Accept (spotlight)